# Reinforcement Learning with Verifiable Rewards: GRPO's Loss, Dynamics, and Success Amplification

## Abstract

Group Relative Policy Optimization (GRPO) was introduced recently and used to train DeepSeek–R1 for promoting reasoning in LLMs under verifiable (binary) rewards. We show that the mean+variance calibration of these rewards induces a contrastive loss in which the contrastive samples are synthetic data drawn from the previous policy. While GRPO was originally paired with clipping to keep updates near the old policy, we analyze variants that differ in reward normalization (mean-only vs. mean+variance) and in how they regularize updates using KL divergence: either penalizing divergence from the previous model (*mirror*), penalizing divergence from a fixed reference model $\pi_{\text{ref}}$, or combining both forms of regularization. For each, the optimal policy $\pi_n$ admits an explicit form in terms of the binary reward and the first and second order statistics of the reward under $\pi_{n-1}$, as well as the policies $\pi_{n-1}$ and $\pi_{\text{ref}}$. Iterating results in a sequence $\{\pi_n\}$ whose *probability of success (PoS)* obeys a simple recurrence that converges to a fixed point determined by the reference PoS and the regularization strength. We further show that this fixed point exceeds the reference, demonstrating that GRPO amplifies the policy's probability of success.

## 1 Introduction

In Reinforcement Learning (RL), a policy is learned by maximizing a reward that encodes constraints or an objective we want the policy to conform to or achieve. Policy gradient methods and actor-critic methods (Sutton and Barto, 1998), enable RL-based training of parametric policies, including Large Language Models (LLMs), particularly when dealing with non-differentiable rewards. Unlike supervised learning or preference optimization, which require labeled training data, reinforcement learning generates *synthetic data* sampled online from the learned policy as training progresses.

Proximal Policy Optimization (PPO), introduced in (Schulman et al., 2017), is a widely used algorithm that facilitates such training. PPO relies on importance sampling from the model's previous ("old") policy while ensuring that updates remain within a certain proximity to the old policy. Policy gradient methods are known for their high variance, and PPO mitigates this by learning a critic that reduces the variance of gradient estimates. The critic normalizes the reward, and PPO's advantage function—defined as the difference between the reward and the critic's evaluation—drives the optimization process.

Group Relative Policy Optimization (GRPO) was recently introduced in DeepSeekMath (Shao et al., 2024). GRPO closely follows PPO's optimization framework but differs in how the advantage is computed. Specifically, GRPO estimates the advantage using Monte Carlo rollouts rather than a learned critic. Additionally, GRPO applies whitening to the advantage function, meaning it standardizes the reward's mean and variance. These statistics are estimated from a "group" of Monte Carlo rollouts corresponding to samples from the LLM policy conditioned on a single input or query to the policy. Whitening the advantage function has been recognized in many PPO implementations as an important ingredient for training stability (Engstrom et al., 2020; Huang et al., 2024).

GRPO therefore eliminates the need for training a separate critic network alongside the LLM policy, instead leveraging efficient sampling from the LLM's policy. This is made feasible by optimized

model serving through VLLM (Kwon et al., 2023). GRPO has been employed in the DeepSeek model series, including DeepSeek-v3 (Liu et al., 2024) and DeepSeek-R1 (Guo et al., 2025). DeepSeek-R1 unlocked reasoning capabilities in open-source models, and its success can be attributed to several factors and innovations, among them: (1) A strong pre-trained model (DeepSeek-v3), (2) The reasoning chain of thoughts `<think>...<think> <answer>...<answer>` and (3) The use of verifiable binary rewards with GRPO to fine-tune the models on reasoning and math tasks.

We focus in this paper on the theoretical grounding of Reinforcement Learning with Verifiable Rewards (RLVR) using GRPO. Verifiable rewards for RL with LLMs (Lambert et al., 2024) typically include (i) correctness checks via string matching to a gold answer when available or via an LLM-as-judge otherwise (Guo et al., 2025; Hugging Face, 2024; Luo et al., 2025; Guan et al., 2025). Additionally, (ii) execution-based pass/fail in code generation (interpreters and unit tests) and (iii) simple binary checks for formatting/refusals provide scalable 0/1 signals for training (Hugging Face, 2024; Guo et al., 2025; Lambert et al., 2024). Verifiable rewards balance simplicity and bias and are thought to be less prone to reward hacking than reward models learned from preference data. We note that a recent paper (Vojnovic and Yun, 2025) studies GRPO with a focus on the policy obtained using an approximation of the KL divergence used in practical implementations.

**Related Work.** Several variants of GRPO have been proposed in the literature. The original GRPO's practical recipe (Shao et al., 2024) combines PPO-style clipping with an explicit KL regularizer to a frozen reference model. On the other hand, mirror-descent style updates that regularize to the *previous* iterate (rather than a fixed reference) have been studied under the Mirror Descent Policy Optimization (MDPO) framework, which interprets each step as approximately solving a trust-region problem via a Bregman (KL) proximity term to $\pi_{n-1}$ (see for example (Schulman et al., 2015; Tomar et al., 2021; Gunter et al., 2024)). "Dr. GRPO" (Liu et al., 2025) is a variant that removes variance normalization (i.e., uses mean-only normalization of group rewards), simplifying the scaling while keeping the same overall training loop. Finally, recent large-scale systems such as DAPO (Yu et al., 2025) report strong results when removing the reference-model KL entirely (i.e., training reference-free), alongside additional engineering choices such as decoupled clipping and dynamic sampling.

Our work aims at understanding the differences between these variants from a theoretical point of view and to explain their empirical success. Our main contributions are:

1. **Contrastive Loss (Sec. 2).** We show that GRPO with calibrated verifiable rewards is equivalent to an *adaptive, weighted contrastive loss* evaluated on samples from the previous policy.

2. **Policy Recursions.** Leveraging this equivalence, we derive, for multiple GRPO variants, a closed-form recursion for the optimal policy as a function of $\pi_{\mathrm{ref}}$, $\pi_{n-1}$, and the previous policy's probability of success (PoS) $p_{n-1}$. Section 3 analyzes GRPO (no clipping) with a KL penalty to the reference; Section 4 studies *Mirror GRPO* with a KL penalty to the previous iterate only; Appendix G covers the mixed (two-KL) case i.e mixed KL penalties to reference and previous iteration; and Section 5 treats the mean-only normalization.

3. **PoS Dynamics , Robustness to Noise & Success Amplification.** We prove that the induced PoS sequence $(p_n)$ satisfies a recursion admitting a fixed point $p^*$ and, under mild assumptions, $p_n \to p^*$ with $p^* \geq p_{\mathrm{ref}}$, establishing *success amplification* for GRPO. The stepwise monotonicity of $(p_n)$ depends on the specific variant. We analyze the robustness of GRPO to noisy rewards in Section 6. The dynamic of the PoS and robustness to noisy rewards are verified empirically in Section 7. Code is provided in supplementary material.

## 2 GRPO WITH VERIFIABLE REWARDS AS AN ADAPTIVE WEIGHTED CONTRASTIVE LOSS

Let $\rho_Q$ be a distribution of prompts or questions, and let $r$ be a reward function that evaluates the output $o \in \mathcal{O}$ of a policy. As discussed in the introduction, we restrict our analysis to verifiable rewards, meaning binary rewards, $r : \mathcal{Q} \times \mathcal{O} \to \{0, 1\}$. Given a prompt $q \sim \rho_Q$, let $\pi_\theta(o|q)$ be the policy of an LLM, where $o$ represents the sequence outcome and $\theta \in \Theta$ the parameters of the model. $\pi_{\theta_{\mathrm{old}}}$ denotes the "old" policy or the policy from a previous iteration. $\pi_{\mathrm{ref}}$ corresponds to the

reference policy, and KL is the Kullback–Leibler divergence :

$$\mathsf{KL}(\pi || \pi_{\text{ref}}) = \mathbb{E}_{q\sim\rho_Q}\mathbb{E}_{o\sim\pi(.|q)} \log\left(\frac{\pi(o|q)}{\pi_{\text{ref}}(o|q)}\right)$$

We note the mean and variance of the reward under a policy $\nu$ as follows:

$$\mu_\nu(q) = \mathbb{E}_{o'\sim\nu(.|q)} r(q, o') \quad \sigma_\nu^2(q) = \mathsf{Var}_{o'\sim\nu(.|q)} r(q, o').$$

For a regularization parameter $\beta > 0$, we start by recalling GRPO's optimization problem (Shao et al., 2024):

$$\max_\theta \mathbb{E}_{q\sim\rho_Q}\mathbb{E}_{o\sim\pi_{\theta_{\text{old}}}(.|q)} f_\epsilon\left(\frac{\pi_\theta(o|q)}{\pi_{\theta_{\text{old}}}(o|q)}, A_{\pi_{\theta_{\text{old}}}}(q, o)\right) - \beta\mathsf{KL}(\pi_\theta || \pi_{\text{ref}}) \qquad \text{(GRPO-Clip)}$$

where the "advantage" for an outcome $o$, $A(q, o)$ is given by the whitened reward:

$$A_{\pi_{\theta_{\text{old}}}}(q, o) = \frac{r(q, o) - \mu_{\pi_{\theta_{\text{old}}}}(q)}{\sigma_{\pi_{\theta_{\text{old}}}}(q)}, \qquad (1)$$

and for $\epsilon \in [0, 1]$, the clipping function $f_\epsilon$ is given by $f_\epsilon(x, y) = \min(xy, \text{clip}(x, 1-\epsilon, 1+\epsilon)y)$.

We see that GRPO optimizes the whitened reward (referred to as advantage, $A(q, o)$, in (Shao et al., 2024)) using importance sampling from the "old" policy while maintaining the optimized policy close to $\pi_{\text{ref}}$ as measured by the KL divergence. The clipping used in equation GRPO-Clip ensures that the likelihood ratio between the policy and the old policy is maintained within $[1-\epsilon, 1+\epsilon]$.

## 2.1 WHITENING THE REWARDS IN GRPO AS MEANS OF CALIBRATION

Recall that our reward $r$ is a verifiable reward that evaluates correctness of a reasoning or code execution, so $r(q, o) \in \{0, 1\}$. We note the probability of success of the old policy $\pi_{\text{old}}$:

$$p(q) = p_{\theta_{\text{old}}}(q) = \mathbb{P}_{o\sim\pi_{\theta_{\text{old}}}(.|q)}(r(q, o) = 1) \qquad (2)$$

Hence, for a Bernoulli random variable, the mean and variance are:

$$\mu_{\pi_{\theta_{\text{old}}}}(q) = p(q) \text{ and } \sigma_{\pi_{\theta_{\text{old}}}}^2(q) = p(q)(1-p(q)).$$

Let us assume in the following that $0 < p(q) < 1$ so that $\sigma_{\pi_{\theta_{\text{old}}}}^2(q) > 0$. Replacing mean and variance in the whitened reward in equation 1 we obtain :

$$A_{\pi_{\theta_{\text{old}}}}(q, o) = \begin{cases} \frac{1-p(q)}{\sqrt{p(q)(1-p(q))}} & \text{if } r(q, o) = 1, \\ -\frac{p(q)}{\sqrt{p(q)(1-p(q))}} & \text{if } r(q, o) = 0. \end{cases} \quad \text{i.e,} \quad A_{\pi_{\theta_{\text{old}}}}(q, o) = \begin{cases} \sqrt{\frac{1-p(q)}{p(q)}} & \text{if } r(q, o) = 1, \\ -\sqrt{\frac{p(q)}{1-p(q)}} & \text{if } r(q, o) = 0. \end{cases}$$
$$(3)$$

**Calibrated reward behavior.** We see that the whitening or the normalization of the verifiable reward in GRPO, calibrates the reward with respect to the conditional distribution of the reward under $\pi_{\theta_{\text{old}}}(.|q)$ for every prompt $q$. This normalization results in a calibration of the reward that involves nonlinear functions of the probability of the success (PoS) of the old policy $p(q)$. See Figure 1 for an illustration. For a correct answer $r(q, o) = 1$, the calibrated reward is positive and *decreases* with the PoS $p(q)$: rare successes (small $p(q)$) receive more credit than easy ones (large $p(q)$). For an incorrect answer ($r(q, o) = 0$), the calibrated reward is negative, and its absolute value is increasing with $p(q)$. Wrong outcomes are more penalized when success is likely (for high $p(q)$) and less penalized when success is rare (low $p(q)$).

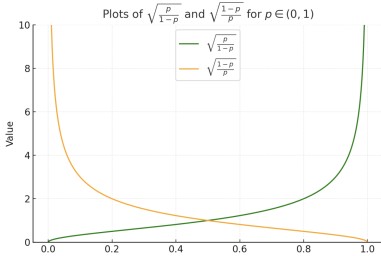

Figure 1: Weighting of GRPO with the probability of success of the old policy.

## 2.2 GRPO WITH VERIFIABLE REWARD AS A WEIGHTED CONTRASTIVE LOSS

Replacing the calibrated reward $A_{\pi_{\theta_{\text{old}}}}(q, o)$ (equation 3) for a verifiable reward in GRPO's optimization objective equation GRPO-Clip we obtain:

$$\mathbb{E}_{o \sim \pi_{\theta_{\text{old}}}(.|q)} f_\epsilon \left( \frac{\pi_\theta(o|q)}{\pi_{\theta_{\text{old}}}(o|q)}, A_{\pi_{\theta_{\text{old}}}}(q, o) \right) = \sqrt{\frac{1 - p(q)}{p(q)}} \, \mathbb{E}_{o \sim \pi_{\theta_{\text{old}}}(.|q), \, r(q,o)=1} \min \left( \frac{\pi_\theta(o|q)}{\pi_{\theta_{\text{old}}}(o|q)}, 1 + \epsilon \right)$$

$$- \sqrt{\frac{p(q)}{1 - p(q)}} \, \mathbb{E}_{o \sim \pi_{\theta_{\text{old}}}(.|q), \, r(q,o)=0} \max \left( \frac{\pi_\theta(o|q)}{\pi_{\theta_{\text{old}}}(o|q)}, 1 - \epsilon \right),$$

where we used that for $x > 0$ and $y > 0$, $f_\epsilon(x, y) = x \min(y, 1 + \epsilon)$ and for $x > 0, y < 0$, $f_\epsilon(x, y) = x \max(y, 1 - \epsilon)$.

The overall cost is further obtained by taking expectation over $q$, noting $p(q) = p_{\theta_{\text{old}}}(q)$:

$$\mathbb{E}_{q \sim \rho_Q} \sqrt{\frac{1 - p_{\theta_{\text{old}}}(q)}{p_{\theta_{\text{old}}}(q)}} \, \mathbb{E}_{o \sim \pi_{\theta_{\text{old}}}(.|q)} \min \left( \frac{\pi_\theta(o|q)}{\pi_{\theta_{\text{old}}}(o|q)}, 1 + \epsilon \right) \mathbb{1}_{r(q,o)=1}$$

$$- \mathbb{E}_{q \sim \rho_Q} \sqrt{\frac{p_{\theta_{\text{old}}}(q)}{(1 - p_{\theta_{\text{old}}}(q))}} \, \mathbb{E}_{o \sim \pi_{\theta_{\text{old}}}(.|q)} \max \left( \frac{\pi_\theta(o|q)}{\pi_{\theta_{\text{old}}}(o|q)}, 1 - \epsilon \right) \mathbb{1}_{r(q,o)=0} - \beta \mathsf{KL}(\pi_\theta || \pi_{\text{ref}})$$

We see that GRPO is effectively a weighted contrastive loss that is weighted by a ratio depending on the probability of success of $\pi_{\theta_{\text{old}}}(.|q)$. We see from the weights plots that if the success probability of the old policy is high ($p > 0.5$), the weighting for points with success is low since the old policy is already good, and for failing points the weight is high and hence they are more penalized. On the other hand if the success probability of old policy is low ($p < 0.5$), the weighting for points with success is high since we want to reinforce those successes, and for failing points these are still penalized but with a small weight.

## 2.3 STABILIZED GRPO

Note that in the previous sections we assumed that $0 < p(q) < 1$, so we ensure $\sigma^2_{\pi_{\theta_{\text{old}}}}(q) > 0$. In the following, we alleviate this by adding a smoothing factor $\varepsilon \in (0, 1]$ in the advantage as follows:

$$A_{\pi_{\theta_{\text{old}}}}(q, o) = \frac{r(q, o) - \mu_{\pi_{\theta_{\text{old}}}}(q)}{\sqrt{\sigma^2_{\pi_{\theta_{\text{old}}}}(q) + \varepsilon}}.$$

This results in the following stabilized whitened reward with a smoothing $\varepsilon > 0$.

$$A_{\pi_{\theta_{\text{old}}}}(q, o) = \begin{cases} +\omega_\varepsilon^+(p(q)), & r(q, o) = 1, \\ -\omega_\varepsilon^-(p(q)), & r(q, o) = 0, \end{cases} \quad \omega_\varepsilon^+(p) = \frac{1 - p}{\sqrt{p(1-p) + \varepsilon}}, \quad \omega_\varepsilon^-(p) = \frac{p}{\sqrt{p(1-p) + \varepsilon}}, \tag{4}$$

Replacing the stabilized advantage in equation GRPO-Clip, we obtain the following contrastive optimization:

$$\max_\theta \mathbb{E}_{q \sim \rho_Q} \left\{ \omega_\varepsilon^+(p_{\theta_{\text{old}}}(q)) \mathbb{E}_{o \sim \pi_{\theta_{\text{old}}}(.|q)} \min \left( \frac{\pi_\theta(o|q)}{\pi_{\theta_{\text{old}}}(o|q)}, 1 + \epsilon \right) \mathbb{1}_{r(q,o)=1} \right.$$

$$\left. - \omega_\varepsilon^-(p_{\theta_{\text{old}}}(q)) \mathbb{E}_{o \sim \pi_{\theta_{\text{old}}}(.|q)} \max \left( \frac{\pi_\theta(o|q)}{\pi_{\theta_{\text{old}}}(o|q)}, 1 - \epsilon \right) \mathbb{1}_{r(q,o)=0} \right\} - \beta \mathsf{KL}(\pi_\theta || \pi_{\text{ref}})$$

**Stabilized GRPO with No Clipping** To simplify Equation GRPO-Clip, let us consider this objective without the clipping ($\epsilon \to +\infty$); we obtain:

$$\max_\theta \mathbb{E}_{q \sim \rho_Q} \mathbb{E}_{o \sim \pi_{\theta_{\text{old}}}(.|q)} \frac{\pi_\theta(o|q)}{\pi_{\theta_{\text{old}}}(o|q)} A_{\pi_{\theta_{\text{old}}}}(q, o) - \beta \mathsf{KL}(\pi_\theta || \pi_{\text{ref}}) \tag{GRPO}$$

Taking the clipping parameter $\epsilon \to \infty$ we obtain GRPO with no clipping equivalent contrastive optimization as follows:

$$
\max_\theta \mathbb{E}_{q \sim \rho_\mathcal{Q}} \Big\{ \omega_\varepsilon^+ (p_{\theta_{\text{old}}}(q)) \mathbb{E}_{o \sim \pi_{\theta_{\text{old}}}(.|q)} \frac{\pi_\theta(o|q)}{\pi_{\theta_{\text{old}}}(o|q)} \mathbb{1}_{r(q,o)=1}
$$

$$
- \omega_\varepsilon^- (p_{\theta_{\text{old}}}(q)) \mathbb{E}_{o \sim \pi_{\theta_{\text{old}}}(.|q)} \frac{\pi_\theta(o|q)}{\pi_{\theta_{\text{old}}}(o|q)} \mathbb{1}_{r(q,o)=0} \Big\} - \beta \text{KL}(\pi_\theta || \pi_{\text{ref}}) \qquad \text{(GRPO-No-Clip)}
$$

which is equivalent to the following problem:

$$
\max_\theta \mathbb{E}_{q \sim \rho_\mathcal{Q}} \Big\{ \omega_\varepsilon^+ (p_{\theta_{\text{old}}}(q)) \mathbb{E}_{o \sim \pi_\theta(.|q)} \mathbb{1}_{r(q,o)=1} - \omega_\varepsilon^- (p_{\theta_{\text{old}}}(q)) \mathbb{E}_{o \sim \pi_\theta(.|q)} \mathbb{1}_{r(q,o)=0} \Big\} - \beta \text{KL}(\pi_\theta || \pi_{\text{ref}}),
$$
$$(5)$$

We will focus first on this non-clipped version.

### 2.4 GRPO ITERATIONS

Algorithm 1 in Appendix C summarizes GRPO iterations (Stabilized and no clipping). We see that GRPO iterations can be written as a sequence of optimization resulting in policies we denote by $\pi_{\theta_n}$ the policy at iteration $n$. We see that GRPO iterations can be written for $n \geq 1$:

$$
\theta_n = \arg \max_\theta \mathbb{E}_{q \sim \rho_\mathcal{Q}} \Big\{ \omega_\varepsilon^+ (p_{\theta_{n-1}}(q)) \mathbb{E}_{o \sim \pi_\theta(.|q)} \mathbb{1}_{r(q,o)=1} - \omega_\varepsilon^- (p_{\theta_{n-1}}(q)) \mathbb{E}_{o \sim \pi_\theta(.|q)} \mathbb{1}_{r(q,o)=0} \Big\} - \beta \text{KL}(\pi_\theta || \pi_{\text{ref}}),
$$
$$(6)$$

Note that in Algorithm 1, expectations are estimated using importance sampling from $\pi_{\theta_{n-1}}$, and each maximization problem is solved via gradient for $\mu$ steps. PoS are estimated using a group size $G$, i.e $G$ Monte-Carlo rollouts from $\pi_{\theta_{\text{old}}}(.|q)$.

In the following we will replace the maximization on the parameter space of the policy by maximizing over the space of policies (i.e optimization on the probability space) in order to analyze the dynamics of GRPO iterations as follows, for $n \geq 1$:

$$
\pi_n = \arg \max_\pi \mathbb{E}_{q \sim \rho_\mathcal{Q}} \Big\{ \omega_\varepsilon^+ (p_{n-1}(q)) \mathbb{E}_{o \sim \pi(.|q)} \mathbb{1}_{r(q,o)=1} - \omega_\varepsilon^- (p_{n-1}(q)) \mathbb{E}_{o \sim \pi(.|q)} \mathbb{1}_{r(q,o)=0} \Big\} - \beta \text{KL}(\pi || \pi_{\text{ref}}),
$$
$$\text{(GRPO Iterations)}$$

where $p_{n-1}(q)$ is the probability of success of the policy $\pi_{n-1}(\cdot|q)$:

$$
p_{n-1}(q) = \mathbb{E}_{o \sim \pi_{n-1}(.|q)} \mathbb{1}_{r(q,o)=1} \qquad (7)
$$

and the weights $\omega_\varepsilon^+$ and $\omega_\varepsilon^-$ are given in equation 4. We assume all throughout the paper that $\pi_0 = \pi_{\text{ref}}$. Note that moving the optimization from a parametric space to the probability space can be seen as assuming that the hypothesis class of the parametric policies is large enough to represent all policies. Note that in GRPO iterations the policy at iteration $n$ depends upon the policy $\pi_{n-1}$ via the probability of success $p_{n-1}$, as well on the reference policy via the KL regularizer.

## 3 GRPO DYNAMICS: FIXED POINT ITERATION FOR PROBABILITY OF SUCCESS

Our goal in this Section is to analyze the dynamics of the GRPO iterations given in equation GRPO Iterations.

**Theorem 1** (GRPO Policy Dynamics). *Optimal GRPO iterations policies solving equation GRPO Iterations satisfy the following recursion, for $n \geq 1$:*

$$
\pi_n(o|q) = \frac{1}{Z_{n-1}(q)} \pi_{\text{ref}}(o|q) \exp \left( \frac{1}{\beta} \left( \omega_\varepsilon^+ (p_{n-1}(q)) \mathbb{1}_{r(q,o)=1} - \omega_\varepsilon^- (p_{n-1}(q)) \mathbb{1}_{r(q,o)=0} \right) \right),
$$

where $Z_{n-1}(q) = p_{\text{ref}}(q) \exp\left(\frac{1}{\beta}\omega_\varepsilon^+(p_{n-1}(q))\right) + (1-p_{\text{ref}}(q)) \exp\left(-\frac{1}{\beta}\omega_\varepsilon^-(p_{n-1}(q))\right)$, *where the weights* $\omega_\varepsilon^+$ *and* $\omega_\varepsilon^-$ *are given in equation 13, the probability of success* $p_{n-1}(q)$ *of policy* $\pi_{n-1}(\cdot|q)$ *is given in equation 7, and* $p_{\text{ref}}(q)$ *is the probability of success of the reference policy* $\pi_{\text{ref}}(\cdot|q)$:
$p_{\text{ref}}(q) = \mathbb{E}_{o\sim\pi_{\text{ref}}(\cdot|q)} \mathbb{1}_{r(q,o)=1}$.

We turn now to the recursion satisfied by the probability of success $p_n(q)$ of the policy $\pi_n(\cdot|q)$, we have the following theorem that shows that this success probability satisfies a fixed point iteration:

**Theorem 2** (GRPO's Probability of Success Fixed Point Iteration). *Assume* $p_{\text{ref}} > 0$, *define for* $\beta > 0$:

$$h_{\varepsilon,p_{\text{ref}}}(p) = \frac{1}{1 + \frac{1-p_{\text{ref}}}{p_{\text{ref}}} \exp\left(-\frac{1}{\beta}\frac{1}{\sqrt{p(1-p)+\varepsilon}}\right)}$$

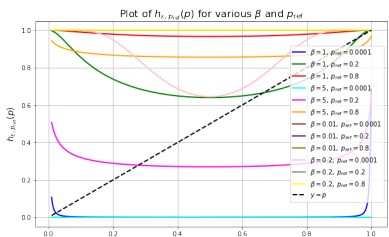

*The probability of success along GRPO's iteration satisfies the following fixed point iteration, i.e, we have almost surely for all q for* $n \geq 1$

$$p_n(q) = h_{\varepsilon,p_{\text{ref}}(q)}(p_{n-1}(q)), \qquad (8)$$

Figure 2: Fixed points as function of $\beta$ and $p_{\text{ref}}$ for $\varepsilon = 1e^{-5}$.

*and* $p_0(q) = p_{\text{ref}}(q)$.

**Remark 1** (Importance of $\varepsilon > 0$). *Note if* $\varepsilon = 0$, $h_{\varepsilon,p_{\text{ref}}}$ *is no longer continuous at* 0 *and* 1 *and we can no longer guarantee existence of fixed points on* $[0,1]$.

We study in the following proposition properties of the function $h_{\varepsilon,p_{\text{ref}}}$:

**Proposition 1** (Properties of $h_{\varepsilon,p_{\text{ref}}}$). $h_{\varepsilon,p_{\text{ref}}}$ *satisfies the following properties:*

- *Existence of fixed points:* $h_{\varepsilon,p_{\text{ref}}}$ *is continuous from* $[0,1]$ *to* $[0,1]$ *and hence admits at least a fixed point* $p^*$ *in* $[0,1]$ *(no guarantees for a unique fixed point)*

- *Monotonicity:* $h'_{\varepsilon,p_{\text{ref}}}(p) = -h_{\varepsilon,p_{\text{ref}}}(p)(1 - h_{\varepsilon,p_{\text{ref}}}(p))\frac{1-2p}{2\beta\,[p(1-p)+\varepsilon]^{3/2}}$

    - *if* $p < \frac{1}{2}$, $h'_{\varepsilon,p_{\text{ref}}}(p) < 0$ *and* $h_{\varepsilon,p_{\text{ref}}}(p)$ *is decreasing*
    - *if* $p > \frac{1}{2}$ $h'_{\varepsilon,p_{\text{ref}}}(p) > 0$ *and* $h_{\varepsilon,p_{\text{ref}}}(p)$ *is increasing*
    - *if* $p = \frac{1}{2}$ $h'_{\varepsilon,p_{\text{ref}}}(p) = 0$ *and* $p = \frac{1}{2}$ *achieves its minimum*

- *Let* $\text{logit}(p) = \log\left(\frac{p}{1-p}\right)$, $\sigma(x) = \frac{1}{1+e^{-x}}$, $((\text{logit}\circ\sigma)(x) = x, (\sigma\circ\text{logit})(p) = p)$. *Define* $\Omega_\varepsilon(p) = \omega_\varepsilon^+(p) + \omega_\varepsilon^-(p) = (p(1-p)+\varepsilon)^{-\frac{1}{2}}$. *We have:*

$$h_{\varepsilon,p_{\text{ref}}}(p) = \sigma\left(\text{logit}\left(p_{\text{ref}}\right) + \frac{\Omega_\varepsilon(p)}{\beta}\right).$$

We drop in the sequel $q$, when referring to the sequence $p_n(q)$, and write for short $p_n$. If the sequence defined in GRPO's probability of success iteration equation 8 converges we have therefore by continuity of $h_{\varepsilon,p_{\text{ref}}}$:

$$p_\infty = \lim_{n\to\infty} p_n = \lim_{n\to\infty} h_{\varepsilon,p_{\text{ref}}}(p_{n-1}) = h_{\varepsilon,p_{\text{ref}}}(\lim_{n\to\infty} p_{n-1}) = h_{\varepsilon,p_{\text{ref}}}(p_\infty),$$

and hence $p_\infty = h_{\varepsilon,p_{\text{ref}}}(p_\infty)$, and the limit point probability of success of GRPO $p_\infty = p^*$ is a fixed point of $h_{\varepsilon,p}$ (fixed points exist by virtue of proposition 1). Note that the fixed point $p^*$ is indeed a function of $q$, and this dependency in $h_{\varepsilon,p_{\text{ref}}}$ is via $p_{\text{ref}}(q)$.

We see in Figure 2 various plots of the function $h_{\varepsilon,p_{\text{ref}}}$ for different values of $\beta$ and initialization $p_{\text{ref}}$, as well as the plot of the function $y = p$. Fixed points correspond to the intersections of this line with the curve of $h_{\varepsilon,p_{\text{ref}}}$. We see that the fixed points are not unique in general, and $p^* = 1$ is almost always a fixed point.

### 3.1 GRPO: FIXED POINT ITERATION AND SUCCESS AMPLIFICATION

Note that from the third item in proposition 1 the PoS recurrence in Theorem 2 can be written in terms of success odds as follows:

$$\text{logit}\left(p_n(q)\right) = \text{logit}\left(p_{\text{ref}}(q)\right) + \frac{\Omega_\varepsilon(p_{n-1}(q))}{\beta}$$

**Theorem 3** (GRPO amplifies the probability of success). *For $q \sim \rho_\mathcal{Q}$ assume $0 < p_{\text{ref}}(q) < 1$. Let $p^*(q)$ be a fixed point of $h_{\varepsilon, p_{\text{ref}}(q)}$ we have $p^*(q) > p_{\text{ref}}(q)$.*

We see from Theorem 3 for any prompt $q$, the fixed point PoS $p^*(q)$ of the GRPO iteration leads to an amplification of the probability of success of the reference model $p_{\text{ref}}(q)$. Note if $p_{\text{ref}}(q) = 0$ or $p_{\text{ref}}(q) = 1$, the iteration will lead to $p^*(q) = 0$ and $p^*(q) = 1$ respectively. In this case the fixed point is not necessarily stable and a condition on $\beta$ is needed for its stability (See appendix E.2 )

## 4 MIRROR GRPO: MIRROR DESCENT WITH GRPO CALIBRATED REWARD

Note that we previously considered GRPO with no-clipping and with a KL regularization to $\pi_{\text{ref}}$.

We consider here a mirror GRPO with a regularization to $\pi_{n-1}$ in addition to $\pi_{\text{ref}}$. For $n \geq 1$:

$$\max_\pi \ \mathbb{E}_{q \sim \rho_\mathcal{Q}} \left( \mathbb{E}_{\pi(\cdot|q)} A_{n-1}(q, \cdot) - \beta \left( \alpha \mathsf{KL}\Big(\pi(\cdot \mid q) \,\Big\|\, \pi_{\text{ref}}(\cdot \mid q)\Big) + (1 - \alpha)\mathsf{KL}\Big(\pi(\cdot \mid q) \,\Big\|\, \pi_{n-1}(\cdot \mid q)\Big)\right)\right),$$

(9)

where:

$$A_{n-1}(q, o) = \begin{cases} +\omega_\varepsilon^+(p_{n-1}(q)), & r(q, o) = 1 \\ -\omega_\varepsilon^-(p_{n-1}(q)), & r(q, o) = 0, \end{cases}$$

(10)

and $p_{n-1}(q) = \mathbb{P}_{\pi_{n-1}(\cdot|q)}(r(q, o) = 1)$, and $\pi_0 = \pi_{\text{ref}}$.

If $\alpha = 1$ we obtain KL regularization to the $\pi_{\text{ref}}$. If $\alpha = 0$, we obtain mirror regularization to the previous iteration without considering the reference. Many recent works suggested using $\alpha = 0$ such as DAPO (Yu et al., 2025) i.e removing the regularization to the reference in GRPO while maintaining the clipping. Note that proximal methods with regularization to previous iterates play the same role of clipping (Tomar et al., 2021; Gunter et al., 2024). Indeed PPO style clipping (Schulman et al., 2017) was introduced as an approximation of such proximal mirror descent.

We study in the following the case $\alpha = 0$, the general case $\alpha > 0$ is analyzed in Appendix G.

Theorem 4 gives the optimal policy for Mirror-GRPO iterations, and its corresponding PoS recurrence:

**Theorem 4** (Mirror-GRPO, $\alpha = 0$). *Fix $\alpha = 0$ and a prompt $q$ and let $\beta > 0$. Let $\Omega_\varepsilon(p) = \frac{1}{\sqrt{p(1-p)+\varepsilon}}$. Then the following holds:*

*1.* Optimal policy. *The maximizer of equation 9 is*

$$\pi_n(o|q) = \frac{1}{Z_{n-1}(q)} \pi_{n-1}(o|q) \exp\left(\frac{1}{\beta}\left(\omega_\varepsilon^+(p_{n-1}(q))\mathbb{1}_{r(q,o)=1} - \omega_\varepsilon^-(p_{n-1}(q))\mathbb{1}_{r(q,o)=0}\right)\right),$$

*where $Z_{n-1}(q) = p_{n-1}(q)\exp\left(\frac{1}{\beta}\omega_\varepsilon^+(p_{n-1}(q))\right) + (1 - p_{n-1}(q))\exp\left(-\frac{1}{\beta}\omega_\varepsilon^-(p_{n-1}(q))\right).$*

*2.* PoS and odds recurrences. *The PoS of $\pi_n(\cdot|q)$, $p_n(q)$, satisfies the following recurrence:*

$$\text{logit}(p_n(q)) = \text{logit}(p_{n-1}(q)) + \frac{\Omega_\varepsilon(p_{n-1}(q))}{\beta},$$

(11)

$$p_n(q) = h_{\varepsilon,\beta}(p_{n-1}(q)) = \sigma\Big(\text{logit}(p_{n-1}(q)) + \Omega_\varepsilon(p_{n-1}(q))/\beta\Big).$$

(12)

When compared with Theorem 2, we see that $p_{n-1}(q)$, replaces $p_{\text{ref}}(q)$ in the logit inside the sigmoid.

**Theorem 5** (Monotone Improvement and Absorbing Fixed Points). *Fix a prompt q, the PoS iterations $p_n(q)$ of Mirror-GRPO ($\alpha = 0$) have the following properties:*

1. ***Monotone improvement and absence of interior fixed points.*** *For any $p_{n-1} \in (0,1)$, $\Omega_\varepsilon(p_{n-1})/\beta > 0$, hence $\text{logit}(p_n) > \text{logit}(p_{n-1})$ and $p_n > p_{n-1}$. Consequently, the equation $p = \sigma(\text{logit}(p) + \Omega_\varepsilon(p)/\beta)$ has no solution in $(0,1)$. The only fixed points are at the boundary: $p \in \{0,1\}$.*

2. ***Convergence and stability.*** *The fixed points of Mirror-GRPO iterations ($\alpha = 0$) satisfy:*

   (a) *If $p_{\text{ref}}(q) \in (0,1)$, then $(p_n(q))_n$ is strictly increasing and bounded by 1, hence $p_n \uparrow 1$.*

   (b) *If $p_{\text{ref}}(q) \in (0,1)$, $p^* = 1$ is (globally) stable fixed point: $\lim_{n\to\infty} p_n(q) = 1$.*

   (c) *If $p_{\text{ref}}(q) = 0$ then $p_n(q) = 0$ for all n.*

When compared with Theorem 3, we see for non-zero $p_{\text{ref}}(q)$, Mirror-GRPO iterations of probability of success converges to 1 that is a stable fixed point, whereas for GRPO with only reference regularization we may have an interior fixed point $p^*(q) > p_{\text{ref}}(q)$. In both cases, for zero $p_{\text{ref}}(q)$, GRPO with reference regularization or Mirror GRPO don't create successes, and the fixed point success remains at zero. From a practical point of view removing the reference regularization is convenient as one does not need to keep in memory the reference model in addition to the current model. In addition it has more favorable PoS guarantees than reference regularization only. Nevertheless in many situations one wants to achieve good performance on a task via RL training while maintaining the knowledge of the reference model and hence the case $\alpha > 0$ is also of interest, we study this case fully in Appendix G. The main takeaway in that scenario where we interpolate between $\alpha = 0$ and $\alpha = 1$, is that we lose monotonic improvement. The PoS iteration incurs what we call a Rényi correction that encodes the mismatches in success and failures between the reference and the previous iteration and we are back to an interior fixed point in $(0,1)$ and no guarantees of global stability as in the mirror-GRPO case.

## 5   Dr. GRPO and mean-only Normalization

We turn now to another reward normalization proposed in Dr. GRPO (Liu et al., 2025). Liu et al. (2025) suggests to use a mean-only normalization in GRPO. In our notations this corresponds to the following reward calibration:

$$A_{\pi_{\theta_{\text{old}}}}(q,o) = \begin{cases} +\omega^+(p(q)), & r(q,o) = 1, \\ -\omega^-(p(q)), & r(q,o) = 0, \end{cases} \quad \omega^+(p) = 1 - p, \quad \omega^-(p) = p. \quad (13)$$

This results in the following (no clipping) Dr. GRPO iterations for PoS:

$$\text{logit}(p_n(q)) = \text{logit}(p_{\text{ref}}(q)) + \frac{1}{\beta}.$$

and the following for Mirror Dr. GRPO

$$\text{logit}(p_n(q)) = \text{logit}(p_{n-1}(q)) + \frac{1}{\beta}.$$

These expressions can be obtained applying Theorem 8 in Appendix F for this particular weighting with $\Omega(p) = 1$. When Compared with (no) clip GRPO, DR. GRPO has a trivial constant fixed point $p^*(q) = \sigma(\text{logit}(p_{\text{ref}}(q)) + \frac{1}{\beta})$. While for Mirror Dr. GRPO , $L_n(q) = \text{logit}(p_n(q))$ is an arithmetic progression and $L_n(q) = L_{\text{ref}}(q) + \frac{n}{\beta}$ and $p_n(q) \uparrow 1$ for non degenerate $p_{\text{ref}}(q) \in (0,1)$. Comparing to Mirror GRPO we have a similar convergence to a PoS of 1 but the iteration are adaptive in the case of Mirror GRPO:

$$\text{logit}(p_n(q)) = \text{logit}(p_{n-1}(q)) + \frac{\Omega_\varepsilon(p_{n-1}(q))}{\beta} = \text{logit}(p_{n-1}(q)) + \frac{1}{\beta(\sigma_{n-1}^2(q) + \varepsilon)^{\frac{1}{2}}},$$

we can think that the variance normalization corresponds to mean-only normalization with an adaptive effective $\beta_{\text{eff}} = \beta\sqrt{\sigma_{n-1}^2(q) + \varepsilon}$. For low variance we make large increments in the logits of PoS and for high variance, we make smaller increments in the logits of PoS.

## 6 ROBUSTNESS OF GRPO VARIANTS TO NOISY REWARDS

We study in this Section the robustness of GRPO variants to noisy rewards that can be encountered in practice, and robustness would be a desired feature of GRPO. Let $\lambda \in [0, 1]$, and $r$ be the true verifiable reward. We consider the following noise model for the verifiable reward $r$:

$$\tilde{r}(q, o) = \begin{cases} r(q, o), & \text{with probability } \lambda \\ 1 - r(q, o), & \text{with probability } 1 - \lambda, \end{cases} \tag{14}$$

for $\lambda = 1$, we have a noiseless reward and for $0 \leq \lambda < 1$ we have a noisy reward. We show in Appendix I that GRPO with noisy rewards has the following form of recursion for the true PoS:

$$\text{logit}(p_n) = \text{logit}(p_\circ) + \Phi_\lambda\big(\Omega((2\lambda - 1)p_{n-1} + 1 - \lambda)\big), \qquad \Phi_\lambda(\Omega) := \log \frac{\lambda e^\Omega + 1 - \lambda}{(1 - \lambda)e^\Omega + \lambda}, \tag{15}$$

where $\Omega(p) = \frac{1}{\beta}$ for mean only normalization and $\Omega(p) = \frac{1}{\beta\sqrt{p(1-p)+\varepsilon}}$. For Mirror GRPO ($\alpha = 0$ in equation 9 ) the anchor PoS $p_\circ$ is $p_{n-1}$ and for Mixed Ref-Mirror GRPO ($\alpha > 0$ in equation 9), the anchor $p_\circ = \tilde{p}_{n-1}^{(\alpha)}$ defined in Lemma 3 that incorporates the reference probability of success $p_{\text{ref}}$. For $\lambda = 1$, we recover GRPO iterations in the noiseless case derived earlier.

It is important to note that $\Omega > 0$ and hence $\Phi_\lambda(\Omega) > 0$ if $\lambda > \frac{1}{2}$ and $\Phi_\lambda(\Omega) < 0$ if $\lambda < \frac{1}{2}$ (Proof in Appendix I). Hence we see that noisy GRPO amplifies the PoS above $p_\circ$ at each iteration when $\lambda > \frac{1}{2}$ ,which corresponds to small noise scenarios. On the other hand, the PoS deteriorates for large noise corresponding to $\lambda < \frac{1}{2}$.

## 7 EXPERIMENTAL VALIDATION

**Setup** We use the `GSM8K` dataset from Cobbe et al. (2021) (MIT license), and `Qwen/Qwen2.5-0.5B-Instruct` (Apache 2.0 license) by Yang et al. (2024) as the reference policy. We use GRPO implementation in TRL (von Werra et al., 2020b), and train on the training split of `GSM8K` on a node with 8 GPUs (GPU0 for the vLLM server and 7 other GPUs for distributed training). We use a learning rate lr $\in \{1e^{-6}, 5e^{-6}\}$, clipping $\varepsilon \in \{0.2, 0.05\}$ and the KL regularizer $\beta \in \{0.1, 0.4\}$, and $\mu$ in Algorithm 1 is set to $\mu = 1$. Other hyperparameters are given in Appendix K . We use the correctness of the LLM output as a reward.

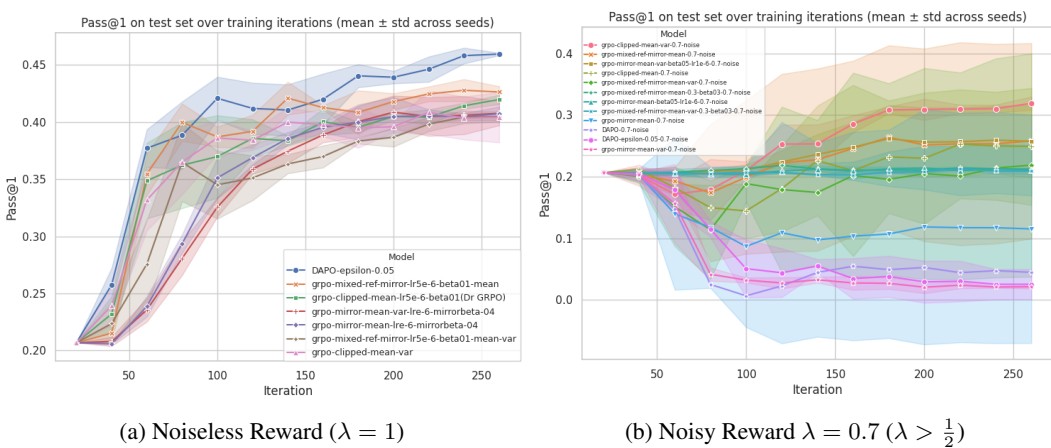

(a) Noiseless Reward ($\lambda = 1$)  (b) Noisy Reward $\lambda = 0.7$ ($\lambda > \frac{1}{2}$)

Figure 3: PoS amplification for GRPO variants for (a) noiseless and (b) noisy settings .

**Success Rate Amplification and Robustness to Noise** We trained different variants of GRPO using three random seeds for each. The variant we considered are : GRPO-CLIPPED-MEAN-VAR that is the original GRPO (Shao et al., 2024) with mean and variance normalization of the reward; GRPO-CLIPPED-MEAN that corresponds to Dr GRPO (Liu et al., 2025) with mean only normalization; MIRROR-GRPO-MEAN and MIRROR-GRPO-MEAN-VAR, corresponding to Mirror GRPO

with mean-variance and mean only normalization respectively; MIXED-REF-MIRROR-GRPO corresponding to mixed KL regularized GRPO to reference and previous policies, with mean-var or mean only normalization of the rewards; as well as DAPO (Yu et al., 2025). Hyper-parameters used for each variant are in the legend of Figure 3a, if unspecified the clipping $\epsilon$ of clipped GRPO variant is 0.2, and learning rate is $5e^{-6}$. Ablations on these hyper-parameters are given in Appendix D Figure 6. We consider two settings for the reward: noiseless setting corresponding to $\lambda = 1$ in equation 14 and the noisy setting with $\lambda = 0.7$ and $\lambda = 0.3$ in equation 14. The success rate of the trained policy with these variants is then evaluated on the test set consisting of 1319 math questions, where for each question the success rate is evaluated using 50 samples, and we report the Pass@1 that corresponds to the average of that success rate on the test dataset.

In the noiseless case, we see in Figure 3a that as predicted by our theory Mirror GRPO and Mixed Ref Mirror GRPO with both mean-var and mean normalization lead to success amplification along the training, and they slightly outperform original GRPO-clipped-mean-var and Dr GRPO (GRPO-clipped-mean). DAPO that is similar in spirit to the mirror GRPO stands out in outperforming all variants. We note that for both mirror GRPO and DAPO as shown in the ablation in Appendix D, Figure 6 a careful hyper-parameter tuning is needed since there is no regularization to the reference model.

In the noisy case, for moderate noise ($\lambda = 0.7$) in Figure 3b (or larger Figure 7 in Appendix), we see that the original GRPO with mean–variance normalization and clipping is the most robust to noise, followed by the Mixed Ref Mirror with mean normalization, and then by Mirror mean–var. We also notice that, in this case, DAPO has a deteriorated performance. For GRPO with clipping and mean–variance normalization, note that it has KL regularization to the reference model. Clipping plays two roles in this scenario: it robustifies the training against adversarial noise and it keeps the policy in the vicinity of the previous one. For the Mixed Ref Mirror, inspecting equation 15, since $\Phi_\lambda > 0$ for $\lambda > \frac{1}{2}$ the logit improvement is taken above an anchor that is a mixture of the reference and the previous policy. In particular, having the reference PoS inside this anchor maintains performance improvement and helps fight off deterioration under noise. For large noise $\lambda = 0.3$ ($\lambda < \frac{1}{2}$), we see in Figure 8, performance deteriorates for all variants, this is consistent with our theoretical results (equation 15) since $\Phi_\lambda < 0$ in this case.

# 8 DISCUSSION AND CONCLUSION

Table 4 in the Appendix summarizes different flavors of GRPO we studied in this paper and their corresponding probability of success iterations. From a practical point of view, our study suggests the following for using GRPO in training LLMs.

---

**Practical Takeaways**

- **Normalization equivalence.** The mean+variance normalization in GRPO is equivalent from PoS point of view to mean-only normalization using an adaptive KL regularization $\beta_{\text{eff}} = \beta\sigma(q)$. One can use either a fixed $\beta$ and get constant increments in log PoS odds via mean-only calibration, or use mean calibration with $\beta_{\text{eff}}$ as a KL regularizer which results in adaptive increments that are equivalent.

- **Mirror versus Clipping and Reference Mixing** Mirror GRPO (KL to previous iteration only ) instead of clipped GRPO guarantees monotonic improvement and convergence to PoS of 1 for non degenerate $p_{\text{ref}}$. Mirror GRPO has the best theoretical and practical guarantees. Adding the reference regularization to this mirror descent results in an internal fixed point and no monotonic improvement is guaranteed. Practically speaking, keeping a reference policy in memory increases bandwidth/latency and can slow training for large models.

- **Coverage and exploration.** In all cases GRPO does not create successes and 0 is an absorbing fixed point if $p_{\text{ref}}(q) = 0$. Hence it is important to maintain successes exploration (e.g., temperature, entropy bonus, or data mixing) so successes have nonzero support.

- **Robustness to Noisy Rewards.** For moderate noise levels of rewards, GRPO with clipping, mean-var normalization and reference regularization, as well as Mixed Ref Mirror GRPO offer some robustness to noise.

---

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

## A  LIMITATIONS & OUTLOOK

Our work analyzes GRPO and focuses on verifiable binary reward in a comprehensive way along multiple variants and studies its robustness to noise. Verifiable rewards are important for reasoning and code generation in LLMs. Other rewards that are continuous are not covered in our analysis, one can turn continuous reward to binary via thresholding, nevertheless extending our analysis, techniques and tools introduced in this paper to that setting remains interesting and a venue for future research. Our analysis hinges on KL regularization to previous iterations and does not incorporate clipping. Clipping was introduced as an approximation to the KL regularization to the previous iterations. Our experiments suggest that clipping ensures further robustness to noisy rewards. Understanding clipping within a robust optimization framework is an interesting open question and perhaps introducing a soft differentiable clipping will allow the analysis of the clipping framework, using the techniques introduced in this paper.

## B  SUMMARY

| Method | Loss normalization | PoS recurrence (log-odds) | Fixed point |
|---|---|---|---|
| **No-clip GRPO** (reference, variance-normalized) | mean + variance | $\operatorname{logit} p_n = \operatorname{logit} p_{\mathrm{ref}} + \dfrac{\Omega_\varepsilon(p_{n-1})}{\beta}$ | Interior (implicit): $p = \sigma\big(\operatorname{logit} p_{\mathrm{ref}} + \dfrac{\Omega_\varepsilon(p)}{\beta}\big)$ |
| **No-clip Dr. GRPO** (reference, mean-only) | mean-only | $\operatorname{logit} p_n = \operatorname{logit} p_{\mathrm{ref}} + \dfrac{1}{\beta}$ | Interior (explicit): $p = \sigma\big(\operatorname{logit} p_{\mathrm{ref}} + \dfrac{1}{\beta}\big)$ |
| **Mirror GRPO** (previous, variance-normalized) | mean + variance | $\operatorname{logit} p_n = \operatorname{logit} p_{n-1} + \dfrac{\Omega_\varepsilon(p_{n-1})}{\beta}$ | Boundary $\{0,1\}$: from non-degenerate starts, $p_n \uparrow 1$ |
| **Mirror Dr. GRPO** (previous, mean-only) | mean-only | $\operatorname{logit} p_n = \operatorname{logit} p_{n-1} + \dfrac{1}{\beta}$ | Boundary $\{0,1\}$: from non-degenerate starts, $p_n \uparrow 1$ |
| **Mirror + Reference** (two-KL, variance-normalized) | mean + variance | $\operatorname{logit} p_n = \alpha \operatorname{logit} p_{\mathrm{ref}} + (1-\alpha)\operatorname{logit} p_{n-1}$ $+ \Delta_R(p_{n-1}) + \dfrac{\Omega_\varepsilon(p_{n-1})}{\beta}$ | Interior (implicit): $\operatorname{logit} p = \operatorname{logit} p_{\mathrm{ref}} + \dfrac{\Delta_R(p)}{\alpha}$ $+ \dfrac{\Omega_\varepsilon(p)}{\alpha\beta}$ |
| **Mirror + Reference (Dr.)** (two-KL, mean-only) | mean-only | $\operatorname{logit} p_n = \alpha \operatorname{logit} p_{\mathrm{ref}} + (1-\alpha)\operatorname{logit} p_{n-1}$ $+ \Delta_R(p_{n-1}) + \dfrac{1}{\beta}$ | Interior (implicit): $\operatorname{logit} p = \operatorname{logit} p_{\mathrm{ref}} + \dfrac{\Delta_R(p)}{\alpha}$ $+ \dfrac{1}{\alpha\beta}$ |

$\operatorname{logit}(p)$ is the log-odds; $\Omega_\varepsilon(p) = \dfrac{1}{\sqrt{p(1-p)+\varepsilon}}$; $\beta > 0$; $\alpha \in [0,1]$; and $\Delta_R(\cdot)$ is the Rényi-based bias term (depends on the current policy, hence on $p$).

Figure 4: GRPO variants with fixed $\beta$ and mixed penalty $\beta\big[\alpha \operatorname{KL}(\pi\|\pi_{\mathrm{ref}}) + (1-\alpha)\operatorname{KL}(\pi\|\pi_{n-1})\big]$.

The main dimensions these variants in Table 4 differ on are: 1) the reward calibration: mean and variance normalization as in the original GRPO or mean-only normalization as in Dr GRPO (Liu et al., 2025). Our theory showed that the normalization results in different weighting schemes, nonlinear in the PoS for GRPO and linear in the PoS for Dr GRPO. 2) As discussed earlier the analysis of the PPO style clipping to maintain the policy updates in the vicinity of the old policy is challenging and it has been shown to be more stable to use mirror policy descent to train LLMs with RL (Gunter et al., 2024). Hence we distinguish GRPO variants also with respect to the anchor distribution on which the KL regularization is applied : no-clip refers to $\pi_{\mathrm{ref}}$ regularization only. Mirror corresponds to the KL regularization given in equation 9 with respect to the previous iterate ($\alpha = 0$), we also consider the regularization to both reference and previous iteration (two-KL) for $\alpha > 0$. For $\alpha = 0$, we see that we obtain a monotonic improvement in the PoS whereas mixing the reference and the previous iterate in the iterations does not guarantee monotonic improvement. The PoS iteration in this case depends on the mismatch in success and failures between the reference and the previous iteration that we quantify in Appendix G via a Rényi correction.

---

**Algorithm 1** Iterative GRPO with verifiable rewards, modified from (Shao et al., 2024)

---

1: **Input** initial policy model $\pi_{\theta_{\text{init}}}$; verifiable reward $r$; task prompts $\mathcal{D}$; hyperparameters $\epsilon$, $\beta$, $\mu$
2: policy model $\pi_\theta \leftarrow \pi_{\theta_{\text{init}}}$
3: **for** $n = 1, \dots, M$ **do**
4:     Sample a batch $\mathcal{D}_b$ from $\rho_{\mathcal{Q}}$
5:     Update the old policy model $\pi_{\theta_{\text{old}}} \leftarrow \pi_\theta$
6:     Sample $G$ outputs $\{o_i\}_{i=1}^G \sim \pi_{\theta_{\text{old}}}(\cdot \mid q)$ for each question $q \in \mathcal{D}_b$
7:     Compute rewards $\{r_i\}_{i=1}^G$ for each sampled output $o_i$ by running verifiable reward $r$
8:     Compute $\hat{A}(q, o_i)$ using equation 4, where $\hat{p}(q) = \hat{p}_{\theta_{\text{old}}}(q) = \frac{1}{G} \sum_{i=1}^G \mathbb{1}_{r(q,o_i)=1}$
9:     **for** GRPO iteration = 1, ..., $\mu$ **do**
10:         Update the policy model $\pi_\theta$ by maximizing GRPO objective with gradient ascent
11:     **end for**
12: **end for**
13: **Output** $\pi_\theta$

---

## C  ALGORITHM

## D  ADDITIONAL PLOTS AND EXPERIMENTS

**Trajectory of Success rates Along GRPO Iterations** We randomly select few prompts from GSM8K test set and plot in Figure 5 the trajectory of the success rate of the model along the GRPO iteration (estimated from 50 samples from the model for each prompt). The success rate is computed from checkpoints of the model along the GRPO training. We see that the trajectory of the success rate $p(q)$ resembles the trajectory of a fixed point algorithm (see Figure 10 in Appendix J ). For some points the convergence is fast to the limit point $p^* = 1$, for others we see an oscillatory behavior (similar to the one in last row in Figure 10). Interestingly when $p_{\text{ref}} = 0$, the probability of success does not move much along GRPO iterations as predicted by our theory.

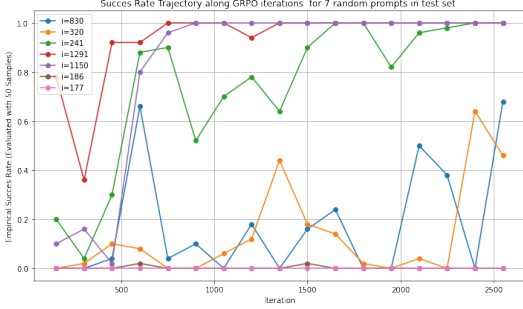

Figure 5: Success rate trajectory of the model on randomly selected prompts along the Clipped GRPO mean-var iters.

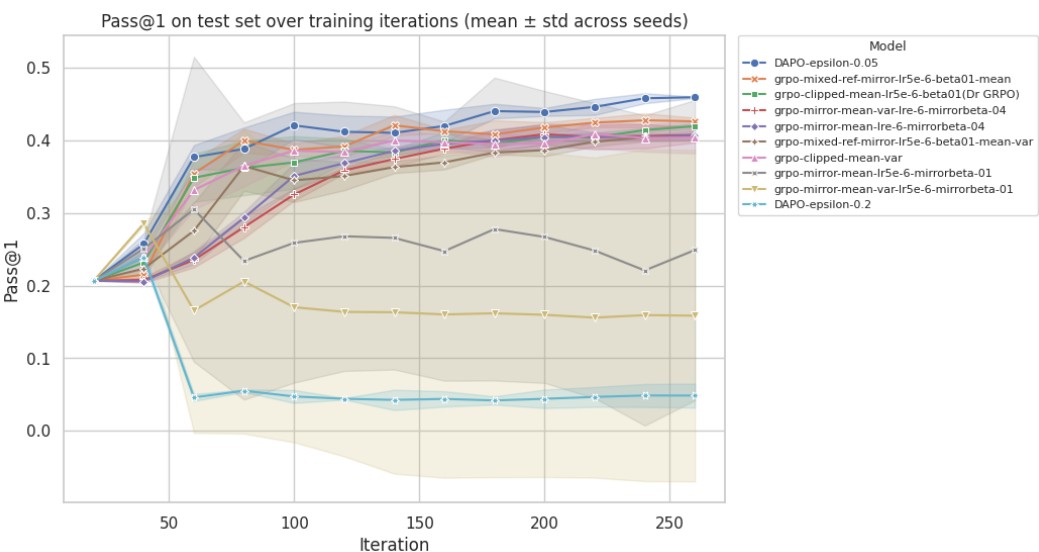

Figure 6: Noiseless Reward: For mirror GRPO that has no regularization to the reference model and only to previous iteration a high regularization parameter and small learning rate are needed for instance $\beta = 0.1$ and a learning rate of $lr = 5e^{-6}$ are not successful, but $\beta = 0.4$ and a learning rate of $lr = 1e^{-6}$ are successful. For DAPO that has no regularization to the reference mode also, a small clipping is needed, $\epsilon = 0.2$ fails whereas $\epsilon = 0.05$ succeeds.

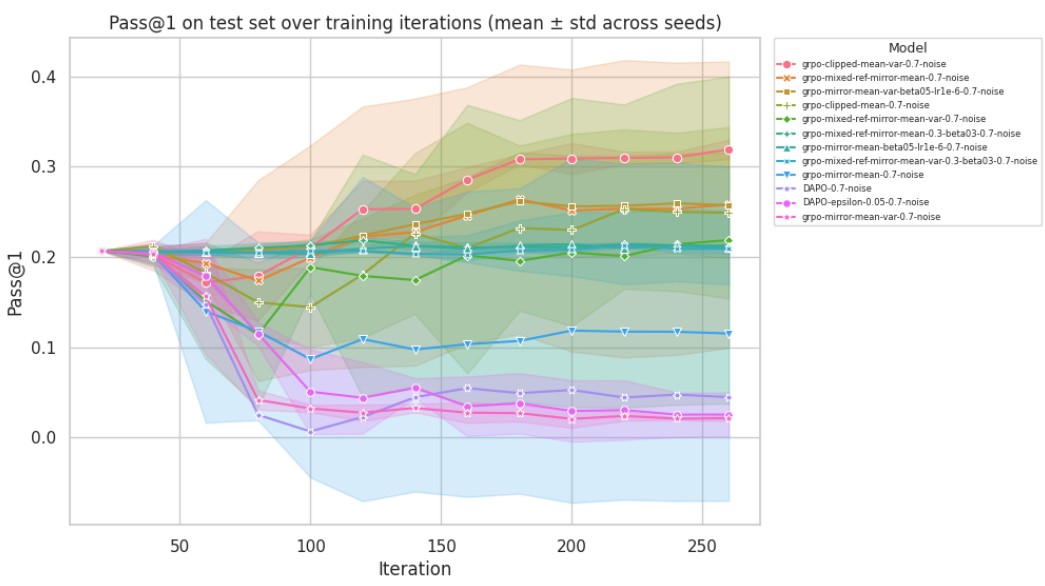

Figure 7: Noisy Reward: noise level $\lambda = 0.7$ ($\lambda > \frac{1}{2}$)

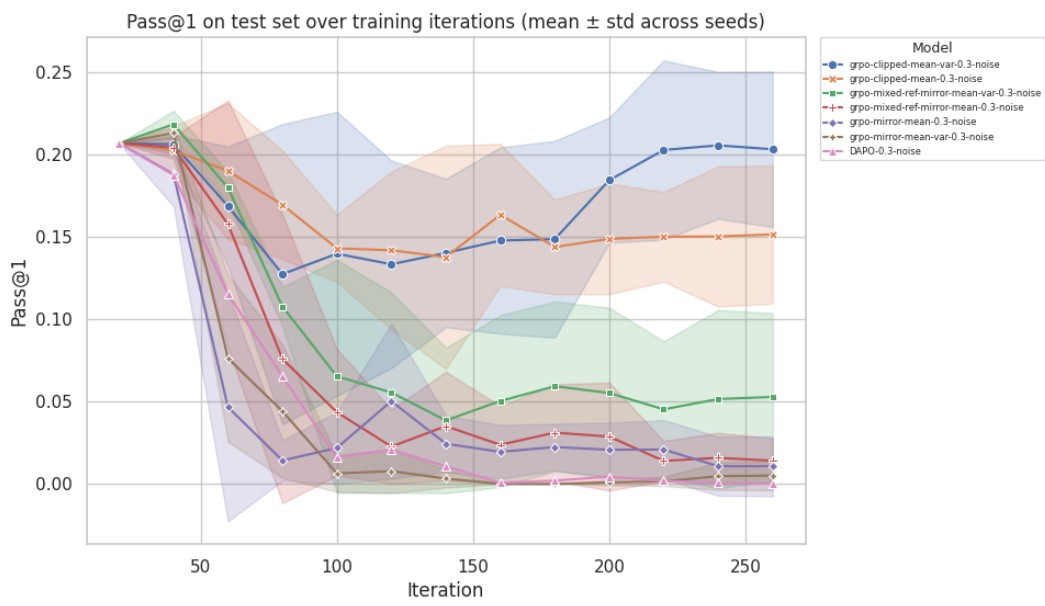

Figure 8: Noisy Reward: noise level $\lambda = 0.3$ ($\lambda < \frac{1}{2}$)

# E (NO-CLIPPING ) GRPO : PROOFS OF SECTION 3

*Proof of Theorem 1.* The objective in Equation equation GRPO Iterations is concave and hence setting the first order optimality conditions (See for example (Mroueh, 2024) ) we obtain:

$$
\pi_n(o|q) = \frac{1}{Z_{n-1}(q)} \pi_{\text{ref}}(o|q) \exp\left( \frac{1}{\beta} \left( \omega_\varepsilon^+(p_{n-1}(q)) \mathbb{1}_{r(q,o)=1} - \omega_\varepsilon^-(p_{n-1}(q)) \mathbb{1}_{r(q,o)=0} \right) \right),
$$

where

$$
\begin{aligned}
Z_{n-1}(q) &= \int d\pi_{\text{ref}}(o|q) \exp\left( \frac{1}{\beta} \left( \omega_\varepsilon^+(p_{n-1}(q)) \mathbb{1}_{r(q,o)=1} - \omega_\varepsilon^-(p_{n-1}(q)) \mathbb{1}_{r(q,o)=0} \right) \right) \\
&= \mathbb{E}_{o \sim \pi_{\text{ref}}(\cdot|q)} \mathbb{1}_{r(q,o)=1} \exp\left( \frac{1}{\beta} \left( \omega_\varepsilon^+(p_{n-1}(q)) \mathbb{1}_{r(q,o)=1} - \omega_\varepsilon^-(p_{n-1}(q)) \mathbb{1}_{r(q,o)=0} \right) \right) \\
&\quad + \mathbb{E}_{o \sim \pi_{\text{ref}}(\cdot|q)} \mathbb{1}_{r(q,o)=0} \exp\left( \frac{1}{\beta} \left( \omega_\varepsilon^+(p_{n-1}(q)) \mathbb{1}_{r(q,o)=1} - \omega_\varepsilon^-(p_{n-1}(q)) \mathbb{1}_{r(q,o)=0} \right) \right) \\
&= \exp\left( \frac{1}{\beta} \omega_\varepsilon^+(p_{n-1}(q)) \right) \mathbb{E}_{o \sim \pi_{\text{ref}}(\cdot|q)} \mathbb{1}_{r(q,o)=1} + \exp\left( -\frac{1}{\beta} \omega_\varepsilon^-(p_{n-1}(q)) \right) \mathbb{E}_{o \sim \pi_{\text{ref}}(\cdot|q)} \mathbb{1}_{r(q,o)=0} \\
&= p_{\text{ref}}(q) \exp\left( \frac{1}{\beta} \omega_\varepsilon^+(p_{n-1}(q)) \right) + (1 - p_{\text{ref}}(q)) \exp\left( -\frac{1}{\beta} \omega_\varepsilon^-(p_{n-1}(q)) \right),
\end{aligned}
$$

where

$$
p_{\text{ref}}(q) = p_0(q) = \mathbb{E}_{o \sim \pi_{\text{ref}}(\cdot|q)} \mathbb{1}_{r(q,o)=1}.
$$

$\square$

*Proof of Theorem 2.* Replacing $\pi_n(\cdot|q)$ by its expression from Theorem 1 we have:

$$p_n(q) = \mathbb{E}_{o\sim\pi_n(.|q)}\mathbb{1}_{r(q,o)=1}$$

$$= \frac{1}{Z_{n-1}(q)}\int d\pi_{\mathrm{ref}}(o|q)\exp\left(\frac{1}{\beta}\left(\omega_\varepsilon^+(p_{n-1}(q))\mathbb{1}_{r(q,o)=1} - \omega_\varepsilon^-(p_{n-1}(q))\mathbb{1}_{r(q,o)=0}\right)\right)\mathbb{1}_{r(q,o)=1}$$

$$= \frac{1}{Z_{n-1}(q)}\exp\left(\frac{1}{\beta}\omega_\varepsilon^+(p_{n-1}(q))\right)\mathbb{E}_{\pi_{\mathrm{ref}}}\mathbb{1}_{r(q,o)=1}$$

$$= \frac{p_{\mathrm{ref}}(q)\exp\left(\frac{1}{\beta}\omega_\varepsilon^+(p_{n-1}(q))\right)}{Z_{n-1}(q)}$$

$$= \frac{p_{\mathrm{ref}}(q)\exp\left(\frac{1}{\beta}\omega_\varepsilon^+(p_{n-1}(q))\right)}{p_{\mathrm{ref}}(q)\exp\left(\frac{1}{\beta}\omega_\varepsilon^+(p_{n-1}(q))\right) + (1-p_{\mathrm{ref}}(q))\exp\left(-\frac{1}{\beta}\omega_\varepsilon^-(p_{n-1}(q))\right)}$$

Replacing the weights expressions from equation 13 we obtain:

$$p_n(q) = \frac{p_{\mathrm{ref}}\exp\left(\frac{1}{\beta}\left(\frac{1-p_{n-1}(q)}{\sqrt{p_{n-1}(q)(1-p_{n-1}(q))+\varepsilon}}\right)\right)}{p_{\mathrm{ref}}\exp\frac{1}{\beta}\left(\frac{1-p_{n-1}(1)}{\sqrt{p_{n-1}(q)(1-p_{n-1}(q))+\varepsilon}}\right) + (1-p_{\mathrm{ref}})\exp\frac{1}{\beta}\left(-\frac{p_{n-1}(q)}{\sqrt{p_{n-1}(q)(1-p_{n-1}(q))+\varepsilon}}\right)}$$

$$(16)$$

Define

$$h_{\varepsilon,p_{\mathrm{ref}}}(p) = \frac{p_{\mathrm{ref}}\exp\left(\frac{1}{\beta}\left(\frac{1-p}{\sqrt{p(1-p)+\varepsilon}}\right)\right)}{p_{\mathrm{ref}}\exp\frac{1}{\beta}\left(\frac{1-p}{\sqrt{p(1-p)+\varepsilon}}\right) + (1-p_{\mathrm{ref}})\exp\frac{1}{\beta}\left(-\frac{p}{\sqrt{p(1-p)+\varepsilon}}\right)}$$

We see therefore that GRPO's probability of success satisfies the following iteration :

$$p_n(q) = h_{\varepsilon,p_{\mathrm{ref}}}(p_{n-1}(q)).$$

We assume here that $0 < p_{\mathrm{ref}} < 1$. We can simplify $h_\varepsilon(p)$ as follows:

$$h_{\varepsilon,p_{\mathrm{ref}}}(p) = \frac{1}{1 + \frac{1-p_{\mathrm{ref}}}{p_{\mathrm{ref}}}\exp\frac{1}{\beta}\left(\frac{-p}{\sqrt{p(1-p)+\varepsilon}} - \frac{1-p}{\sqrt{p(1-p)+\varepsilon}}\right)}$$

$$= \frac{1}{1 + \frac{1-p_{\mathrm{ref}}}{p_{\mathrm{ref}}}\exp\left(-\frac{1}{\beta}\frac{1}{\sqrt{p(1-p)+\varepsilon}}\right)}.$$

$\square$

*Proof of Proposition 1.* **Existence of fixed points** For $\varepsilon > 0$ $h_{\varepsilon,p_{\mathrm{ref}}}$ is continuous function from $[0,1]$ to $[0,1]$ and hence by Brouwer's Fixed Point Theorem at least a fixed point $p^*$ exists in $[0,1]$, i.e $\exists p^* \in [0,1]$ such that $p^* = h_{\varepsilon,p_{\mathrm{ref}}}(p^*)$.

**Monotonicity** Let $\sigma(z) = \frac{1}{1+\exp(-z)}$ and let $A = \frac{1-p_{\mathrm{ref}}}{p_{\mathrm{ref}}}$ and $B(p) = \frac{1}{\beta}\frac{1}{\sqrt{p(1-p)+\varepsilon}}$ hence we have:

$$h_{\varepsilon,p_{\mathrm{ref}}}(p) = \sigma(z(p))$$

where

$$z(p) = -\log(A) + B(p)$$

we have

$$z'(p) = B'(p) = -\frac{1-2p}{2\beta\left[p(1-p)+\varepsilon\right]^{3/2}}$$

Let us compute the derivative :

$$h'_{\varepsilon,p_{\text{ref}}}(p) = \sigma(z(p))(1-\sigma(z(p)))z'(p)$$

$$= -\sigma(z(p))(1-\sigma(z(p)))\frac{1-2p}{2\beta\left[p(1-p)+\varepsilon\right]^{3/2}}$$

- if $p < \frac{1}{2}$, $h'_{\varepsilon,p_{\text{ref}}}(p) < 0$ and $h_{\varepsilon,p_{\text{ref}}}$ is decreasing

- if $p > \frac{1}{2}$ $h'_{\varepsilon,p_{\text{ref}}}(p) > 0$ and $h_{\varepsilon,p_{\text{ref}}}$ is increasing

- if $p = \frac{1}{2}$ $h'_{\varepsilon,p_{\text{ref}}}(p) = 0$

Turning to third point:

$$h_{\varepsilon,p_{\text{ref}}}(p) = \sigma\left(z(p)\right)$$

$$= \sigma\left(\log\frac{p_{\text{ref}}}{1-p_{\text{ref}}} + \frac{1}{\beta}\frac{1}{\sqrt{p(1-p)+\varepsilon}}\right)$$

$$= \sigma\left(\text{logit}(p_{\text{ref}}) + \frac{1}{\beta}\frac{1}{\sqrt{p(1-p)+\varepsilon}}\right)$$

and hence:

$$\text{logit}\left(h_{\varepsilon,p_{\text{ref}}}(p)\right) = \text{logit}(p_{\text{ref}}) + \frac{1}{\beta}\frac{1}{\sqrt{p(1-p)+\varepsilon}}. \tag{17}$$

$\square$

### E.1 PROOFS OF SECTION 3.1

*Proof of Theorem 3.* We claim that any fixed point $p^*$ of $h_\varepsilon$ satisfies

$$p^* > p_{\text{ref}}.$$

We have for all $\beta, \varepsilon > 0$ $\exp\left(-\frac{1}{\beta}\frac{1}{\sqrt{p_{\text{ref}}(1-p_{\text{ref}})+\varepsilon}}\right) < 1$.

$$h_{\varepsilon,p_{\text{ref}}}(p) - p_{\text{ref}} = \frac{1}{1 + \frac{1-p_{\text{ref}}}{p_{\text{ref}}}\exp\left(-\frac{1}{\beta}\frac{1}{\sqrt{p(1-p)+\varepsilon}}\right)} - p_{\text{ref}}$$

$$> \frac{1}{1 + \frac{1-p_{\text{ref}}}{p_{\text{ref}}}} - p_{\text{ref}}$$

$$= p_{\text{ref}} - p_{\text{ref}}$$

$$= 0.$$

Hence for any fixed point we have $h_{\varepsilon,p_{\text{ref}}}(p^*) = p^*$ and we have $p^* > p_{\text{ref}}$. $\square$

### E.2 STABILITY FIXED POINT GRPO WITH REFERENCE ONLY REGULARIZATION

We drop in the sequel the dependency on $q$ to simplify notations and turn to the second question regarding the convergence of the GRPO sequence given in equation 8 to a fixed point $p^*$ of $h_{\varepsilon,p_{\text{ref}}}$. Given the properties of $h_{\varepsilon,p_{\text{ref}}}$, we can characterize the limit point of the GRPO iteration as $n \to \infty$ as follows, as a consequence of the local Banach fixed-point theorem:

**Theorem 6** (Local Fixed Point Convergence). *Let $p^*$ be a fixed point of $h_{\varepsilon, p_{\mathrm{ref}}}$ and assume that have $|h'_{\varepsilon, p_{\mathrm{ref}}}(p^*)| < 1$. Given that $h_{\varepsilon, p_{\mathrm{ref}}}$ and $h'_{\varepsilon, p_{\mathrm{ref}}}$ are continuous in $[0, 1]$, then there exists $\delta > 0$ such the iteration $p_n = h_{\varepsilon, p_{\mathrm{ref}}}(p_{n-1})$ converges to $p^*$, if $p_0 = p_{\mathrm{ref}} \in [p^* - \delta, p^* + \delta]$. In other words under this condition we have:*

$$\lim_{n \to \infty} p_n = p^*.$$

**Lemma 1.** *Let $p^*$ be a fixed point: $p^* = h_{\varepsilon, p_{\mathrm{ref}}}(p^*)$, then we have:*

$$h'_{\varepsilon, p_{\mathrm{ref}}}(p^*) = -h_{\varepsilon, p_{\mathrm{ref}}}(p^*)(1 - h_{\varepsilon, p_{\mathrm{ref}}}(p^*)) \frac{1 - 2p^*}{2\beta \left[ p^*(1 - p^*) + \varepsilon \right]^{3/2}}$$

$$= p^*(1 - p^*) \frac{2p^* - 1}{2\beta \left[ p^*(1 - p^*) + \varepsilon \right]^{3/2}}$$

One condition for local convergence is therefore to have: $|h'_{\varepsilon, p_{\mathrm{ref}}}(p^*)| = p^*(1 - p^*) \frac{|2p^* - 1|}{2\beta \left[ p^*(1 - p^*) + \varepsilon \right]^{3/2}} < 1$ which is satisfied if : $\beta > \mathcal{B}(p^*) = p^*(1 - p^*) \frac{|2p^* - 1|}{2[p^*(1 - p^*) + \varepsilon]^{3/2}}$.

We see from Figure 9 in Appendix J the lower bound on $\beta$ required to ensure local convergence of GRPO iterations to a fixed point $p^*$. Figure 10 in Appendix J shows iteration equation 8 as a function of $n$ for different values of $\beta$ and $p_{\mathrm{ref}}$. We see that in most cases, there is a sharp transition where we observe fast convergence to 1 or to a fixed point $p^*$. For $\beta = 5$ and $p_{\mathrm{ref}} = 0.001$, we see a divergent behavior.

**Remark 2.** *Note that the condition on $\beta$ is stated conditionally on a prompt $q$, to obtain results uniformly on $q$ we need to take $\sup$ on $q$ in all lower bounds.*

**Practical Implications.** In practical implementations GRPO is applied successively in stages where $\pi_{\mathrm{ref}}$ is set to the last iteration from the GRPO training in each stage (Shao et al., 2024). In light of our theory this ensures that we are amplifying the probability of success w.r.t the new $\pi_{\mathrm{ref}}$, coming the previous GRPO stage.

*Proof of Theorem 6.* This is a direct application of local Banach fixed point theorem:

**Theorem 7** (Local Contraction Mapping for One-Dimensional Functions). *Let $f : \mathbb{R} \to \mathbb{R}$ be continuously differentiable, and suppose that $x^* \in \mathbb{R}$ is a fixed point of $f$ (i.e., $f(x^*) = x^*$). Assume that $f'$ is continuous and that*

$$|f'(x^*)| < 1.$$

*Then, by the continuity of $f'$, there exists a radius $r > 0$ and a constant $k$ with*

$$|f'(x)| \le k < 1 \quad \text{for all } x \in [x^* - r, \, x^* + r].$$

*Consequently, $f$ is a contraction on the interval $I = [x^* - r, x^* + r]$, and for any initial guess $x_0 \in I$, the iteration defined by*

$$x_{n+1} = f(x_n)$$

*converges to the unique fixed point $x^*$ in $I$.*

$\square$

# F MIRROR GRPO: PROOF OF SECTION 4

**Theorem 8** (General Theorem with general weights and anchor policy).

$$\pi^* = \mathcal{P}(\nu, \pi_\circ) = \arg\max_\pi \mathbb{E}_{\pi(\cdot|q)} A_\nu(\cdot, q) - \beta \mathsf{KL}(\pi || \pi_\circ)$$

*where*

$$A_\nu(q, o) = \begin{cases} +\omega^+(p_\nu), & r(q, o) = 1, \\ -\omega^-(p_\nu), & r(q, o) = 0, \end{cases} \tag{18}$$

*where $p_\nu = \mathbb{P}_{\nu(\cdot|q)}(r(q, \cdot) = 1)$. Let $\Omega(p) = \omega^+(p) + \omega^-(p)$. The following holds:*

1.

$$\pi^*(o|q) = \frac{\pi_\circ(o|q) \exp \frac{1}{\beta} A_\nu(q,o)}{p_{\pi_\circ}(q) \exp(\frac{1}{\beta}\omega^+(p_\nu(q))) + (1 - p_{\pi_\circ}(q)) \exp(-\frac{1}{\beta}\omega^-(p_\nu(q)))}$$

2. *Let $\pi_{n-1}^\circ(\cdot|q), n \geq 1$ a sequence of anchor probabilities, and $p_{n-1}^\circ(q)$ their corresponding PoS. Let $p_n = p_{\pi_n}$ where $\pi_n$ defined as follows :*

$$\pi_n(q) = \mathcal{P}(\pi_{n-1}(q), \pi_{n-1}^\circ(q)),$$

*we have:*

$$\text{logit}(p_n(q)) = \text{logit}\left(p_{n-1}^\circ(q)\right) + \frac{\Omega(p_{n-1}(q))}{\beta}$$

*and*

$$p_n(q) = \sigma\left(\text{logit}\left(p_{n-1}^\circ(q)\right) + \frac{\Omega(p_{n-1}(q))}{\beta}.\right)$$

*Proof.* The proof of item 1 is the same as in Theorem 1. Turning to the second point we have by taking expectation on success events:

$$
\begin{aligned}
p_*(q) &= \frac{p_{\pi_\circ}(q) \exp(\frac{1}{\beta} w^+(p_\nu(q)))}{p_{\pi_\circ}(q) \exp(\frac{1}{\beta}\omega^+(p_\nu(q))) + (1 - p_{\pi_\circ}(q)) \exp(-\frac{1}{\beta}\omega^-(p_\nu(q)))} \\
&= \frac{1}{1 + \exp(-\text{logit}(p_{\pi_\circ}(q)) - \frac{1}{\beta}\omega^+(p_\nu(q)) - \frac{1}{\beta}\omega^-(p_\nu(q)))} \\
&= \sigma\left(\text{logit}(p_{\pi_\circ}(q)) + \frac{\Omega(p_\nu(q))}{\beta}\right)
\end{aligned}
$$

and hence using that sigmoid and logit are inverse we have:

$$\text{logit}(p_*) = \text{logit}(p_{\pi_\circ}(q)) + \frac{\Omega(p_\nu(q))}{\beta}$$

$\square$

*Proof of Theorem 4.* The theorem is immediate applying Theorem 8 with anchors $\pi_{n-1}$.

$\square$

*Proof of Theorem 4. (1) Monotonicity and no interior fixed points.* Let $L_n = \text{logit}(p_n)$. For $p \in (0,1)$, $\Omega_\varepsilon(p) = 1/\sqrt{p(1-p) + \varepsilon} > 0$, so equation 12 implies $L_n > L_{n-1}$ and hence $p_n > p_{n-1}$. An interior fixed point would solve $L = L + \Omega_\varepsilon(p)/\beta$, impossible since the increment is strictly positive.

It is easy to verify that $p = 0$ and $p = 1$ are fixed points :

$$h_{\varepsilon,\beta}(0) = \sigma(\text{logit}(0) + \frac{1}{\beta\sqrt{\varepsilon}}) = \sigma(-\infty) = 0$$

$$h_{\varepsilon,\beta}(1) = \sigma(\text{logit}(1) + \frac{1}{\beta\sqrt{\varepsilon}}) = \sigma(+\infty) = 1$$

*(2) Convergence and stability.* (1) If $p_0 = p_{\text{ref}} \in (0,1)$, then $(p_n)$ is strictly increasing and bounded by 1, so $p_n \uparrow \bar{p} \leq 1$, and the limit point is $\bar{p} = 1$ the fixed point. (2) the fixed point is unique and stable if $p_{\text{ref}} \in (0,1)$. (3) If $p_{\text{ref}} = 0$, $p_1 = h_{\varepsilon,\beta}(0) = 0$, and so on, zero is an absorbing fixed point.

$\square$

## G   GRPO with Two KL Regularizers: PoS Recursion, and Fixed-Point

Consider the following iteration

$$\pi_n = \arg\max_\pi \; \mathbb{E}_{q\sim\rho_Q} \left( \mathbb{E}_{\pi(\cdot|q)} A_{n-1}(q,\cdot) - \beta \left( \alpha\mathsf{KL}\Big(\pi(\cdot\mid q) \,\Big\|\, \pi_{\mathrm{ref}}(\cdot\mid q)\Big) + (1-\alpha)\mathsf{KL}\Big(\pi(\cdot\mid q) \,\Big\|\, \pi_{n-1}(\cdot\mid q)\Big) \right) \right), \tag{19}$$

**Lemma 2** (Geometric Mean). *For any distributions $\pi, \pi_{\mathrm{ref}}, \pi^\circ$ let $\alpha > 0$*

$$\alpha\mathsf{KL}(\pi\|\pi_{\mathrm{ref}}) + (1-\alpha)\mathsf{KL}(\pi\|\pi^\circ) = \mathsf{KL}(\pi\|\bar\pi^{(\alpha)}) + C(\pi_{\mathrm{ref}}, \pi^\circ),$$

*where $\bar\pi^{(\alpha)} \propto \pi_{\mathrm{ref}}^\alpha \pi^{\circ(1-\alpha)}$ and $C$ is constant in $\pi$.*

*Proof.* See for example (Aminian et al., 2025). □

By Lemma 2, we can rewrite GRPO objective with two KL regularization to previous iteration and to the reference as a single KL regularizer to their geometric mean as follows:

$$\pi_n = \arg\max_\pi \; \mathbb{E}_{q\sim\rho_Q} \left( \mathbb{E}_{\pi(\cdot|q)} A_{n-1}(q,\cdot) - \beta\,\mathsf{KL}(\pi\|\tilde\pi_{n-1}^{(\alpha)}) \right), \tag{20}$$

where

$$\tilde\pi_{n-1}^{(\alpha)} \propto \pi_{\mathrm{ref}}^\alpha \pi_{n-1}^{(1-\alpha)}$$

To apply Theorem 8 we need to have an expression of the PoS under the anchor $\tilde\pi_{n-1}^{(\alpha)}$, as function of $p_{\mathrm{ref}}$ and $p_{n-1}$ so we obtain a recurrence in $p_n$.

Define the following success and failure conditional probabilities:

$$p_{\mathrm{ref},S}(o|q) := \frac{\pi_{\mathrm{ref}}(o\mid q)\,\mathbf{1}_{\{r(q,o)=1\}}}{p_{\mathrm{ref}}(q)}, \quad p_{n-1,S}(o|q) := \frac{\pi_{n-1}(o\mid q)\,\mathbf{1}_{\{r(q,o)=0\}}}{p_{n-1}(q)},$$

and

$$p_{\mathrm{ref},F}(o|q) := \frac{\pi_{\mathrm{ref}}(o\mid q)\,\mathbf{1}_{\{r(q,o)=0\}}}{1-p_{\mathrm{ref}}(q)}, \quad p_{n-1,F}(o|q) := \frac{\pi_{n-1}(o\mid q)\,\mathbf{1}_{\{r(q,o=0)\}}}{1-p_{n-1}(q)}.$$

and let

$$D_\alpha(P\|Q) = \frac{1}{\alpha-1} \log \int p^\alpha q^{(1-\alpha)},$$

be the Rényi divergence of order $\alpha in (0,1)$.

**Lemma 3** (PoS geometric mean). *The probability of success of the geometric mean $\tilde\pi_{n-1}^{(\alpha)}$ satisfies:*

$$\mathrm{logit}\,\tilde p_{n-1}^{(\alpha)} = \alpha\,\mathrm{logit}(p_{\mathrm{ref}}(q)) + (1-\alpha)\,\mathrm{logit}(p_{n-1}(q)) + (\alpha-1)\left(D_\alpha(p_{\mathrm{ref},S}\|p_{n-1,S}) - D_\alpha(p_{\mathrm{ref},F}\|p_{n-1,F})\right)$$

*Proof.* Let $w_S = \int \mathbf{1}_{r(q,o)=1}\tilde\pi_{n-1}^{(\alpha)}$ and $w_F = \int \mathbf{1}_{r(q,o)=0}\tilde\pi_{n-1}^{(\alpha)}$.

$$\tilde p_{n-1}^{(\alpha)} = \frac{\int \mathbf{1}_{r(q,o)=1}\tilde\pi_{n-1}^{(\alpha)}}{\int \mathbf{1}_{r(q,o)=1}\tilde\pi_{n-1}^{(\alpha)} + \int \mathbf{1}_{r(q,o)=0}\tilde\pi_{n-1}^{(\alpha)}} \tag{21}$$

$$= \frac{1}{1+\frac{w_F}{w_S}} \tag{22}$$

$$\frac{w_F}{w_S} = \frac{\int \mathbb{1}_{r(q,o)=0} \pi_{\text{ref}}^{\alpha} \pi_{n-1}^{1-\alpha}}{\int \mathbb{1}_{r(q,o)=1} \pi_{\text{ref}}^{\alpha} \pi_{n-1}^{1-\alpha}}. \tag{23}$$

It is easy to see that :

$$\frac{w_F}{w_S} = \frac{(1 - p_{\text{ref}(q)}^{\alpha})(1 - p_{n-1}(q))^{(1-\alpha)} \int p_{\text{ref},F}^{\alpha}(o|q) p_{n-1,F}^{1-\alpha}(o|q)}{(p_{\text{ref}(q)}^{\alpha})(p_{n-1}(q))^{(1-\alpha)} \int p_{\text{ref},S}^{\alpha}(o|q) p_{n-1,S}^{1-\alpha}(o|q)} \tag{24}$$

Taking log on both sides we have:

$$\log \frac{w_F}{w_S} = \log \frac{(1 - p_{\text{ref}(q)})^{\alpha}}{p_{\text{ref}}^{\alpha}(q)} + \log \left( \frac{(1 - p_{n-1}(q))^{(1-\alpha)}}{p_{n-1}^{1-\alpha}(q)} \right) + \log \int p_{\text{ref},F}^{\alpha}(o|q) p_{n-1,F}^{1-\alpha}(o|q) - \int p_{\text{ref},S}^{\alpha}(o|q) p_{n-1,S}^{1-\alpha}(o|q)$$

$$= -\alpha \operatorname{logit}(p_{\text{ref}}(q)) - (1-\alpha) \operatorname{logit}(p_{n-1}(q)) + (\alpha - 1) \left( D_{\alpha}(p_{\text{ref},F}||p_{n-1,F}) - D_{\alpha}(p_{\text{ref},S}||p_{n-1,S}) \right),$$

where

$$D_{\alpha}(P||Q) = \frac{1}{\alpha - 1} \log \int p^{\alpha} q^{(1-\alpha)},$$

is the Rényi divergence.

Finally we obtain:

$$\tilde{p}_{n-1}^{(\alpha)} = \frac{1}{1 + \exp(-\alpha \operatorname{logit}(p_{\text{ref}}(q)) - (1-\alpha) \operatorname{logit}(p_{n-1}(q)) - (\alpha - 1) \left( D_{\alpha}(p_{\text{ref},S}||p_{n-1,S}) - D_{\alpha}(p_{\text{ref},F}||p_{n-1,F}) \right))}$$

$$= \sigma(\alpha \operatorname{logit}(p_{\text{ref}}(q)) + (1-\alpha) \operatorname{logit}(p_{n-1}(q)) + (\alpha - 1) \left( D_{\alpha}(p_{\text{ref},S}||p_{n-1,S}) - D_{\alpha}(p_{\text{ref},F}||p_{n-1,F}) \right))$$

This gives us finally:

$$\operatorname{logit} \tilde{p}_{n-1}^{(\alpha)} = \alpha \operatorname{logit}(p_{\text{ref}}(q)) + (1-\alpha) \operatorname{logit}(p_{n-1}(q)) + (\alpha - 1) \left( D_{\alpha}(p_{\text{ref},S}||p_{n-1,S}) - D_{\alpha}(p_{\text{ref},F}||p_{n-1,F}) \right).$$

□

**Theorem 9** (PoS recurrence for 2 KL regularizers). *Fix $\alpha \in (0,1), \beta > 0$. The probability of success for the iteration of GRPO with 2 KL regularizer given in equation 19 satisfies the following recurrence:*

$$\operatorname{logit} p_n(q) = \alpha \operatorname{logit}(p_{\text{ref}}(q)) + (1-\alpha) \operatorname{logit}(p_{n-1}(q)) + \underbrace{(1 - \alpha) \left( D_{\alpha}(p_{\text{ref},F}||p_{n-1,F}) - D_{\alpha}(p_{\text{ref},S}||p_{n-1,S}) \right)}_{\Delta_R \text{Rényi Correction}} + \frac{\Omega_{\varepsilon}(p_{n-1})(q)}{\beta}.$$

$$\tag{25}$$

*Proof.* The proof is direct consequence of theorem 8 with geometric mean anchor (as showed in Lemma 2). We replace in theorem 8 the anchor PoS by its expression computed in lemma 3. □

Let $L_n(q) = \operatorname{logit} p_n(q)$ and $L_{\text{ref}}(q) = \operatorname{logit}(p_{\text{ref}}(q))$, hence we have the following recursion:

$$L_n(q) - L_{\text{ref}}(q) = (1-\alpha)(L_{n-1}(q) - L_{\text{ref}}(q)) + (1-\alpha)(D_{\alpha}(p_{\text{ref},F}||p_{n-1,F}) - D_{\alpha}(p_{\text{ref},S}||p_{n-1,S}) + \Omega_{\varepsilon}(p_n)$$

Let us assume that :

$$D_{\alpha}(p_{\text{ref},S}||p_{n-1,S}) \le D_{\alpha}(p_{\text{ref},F}||p_{n-1,F}),$$

i.e conditional successes between reference and previous policy are closer than the failures than we have since $\Omega_{\varepsilon} > 0$:

$$L_n(q) - L_{\text{ref}}(q) \ge (1-\alpha)(L_{n-1}(q) - L_{\text{ref}}(q)) \ge (1-\alpha)^n(L_0 - L_{\text{ref}}) = 0$$

and we obtain that we amplify probability w.r.t to $p_{\text{ref}}$.

# H  BACK TO PARAMETRIC GRPO ITERATIONS

Let $\tilde{\pi}_n = \pi_{\theta_n}$, the sequence of parametric policies solutions of problem equation 6 produced by gradient descent for example as in Algorithm 1. We make the following assumption on the total variation distance TV between these parametric policies and the non-parametric GRPO policies $\pi_n$ given in Theorem 1. We show in this Section if we have approximate policies we can have still asymptotic convergence.

**Assumption 1.** *We assume $\tilde{\pi}_0 = \pi_0 = \pi_{\mathrm{ref}}$ and assume for all $n \geq 1$, there exists $\delta_n \geq 0$ such that:*

$$\mathrm{TV}(\tilde{\pi}_n||\pi_n) \leq \mathrm{TV}(\tilde{\pi}_{n-1}||\pi_{n-1}) + \delta_n,$$

*such that there exists $\delta^* \in [0,1)$ such that $\sum_{i=1}^{n} \delta_i \to \delta^*$ as $n \to \infty$.*

We have the following theorem:

**Theorem 10.** *Under Assumption 1 and assuming that $p_n$ converges to $p^*$ the fixed point of $h_{\varepsilon, p_{\mathrm{ref}}}$. Let $\tilde{p}_n$ the probability of success of the policy $\tilde{\pi}$ we have:*

$$\lim_{n \to \infty} |\tilde{p}_n - p^*| \leq 2\delta^*.$$

*In the case $\delta^* = 0$, we have convergence to the fixed point.*

In Assumption 1 $\delta_n$ represent statistical, approximation and optimization errors. We see from Theorem 10, that as long these error remain small, the probability of success of GRPO parametric policy (estimated from samples and optimized for instance with gradient descent) remains close to the fixed point probability success $p^*$.

*Proof of Theorem 10.* Note that

$$\mathrm{TV}(\tilde{\pi}||\pi) = \frac{1}{2} \sup_{||f||_\infty} \mathbb{E}_{\tilde{\pi}} f - \mathbb{E}_\pi f$$

We have:

$$\begin{aligned}
|\tilde{p}_n - p_n| &= \left| \mathbb{E}_{\tilde{\pi}_n} \mathbb{1}_{r(q,o)=1} - \mathbb{E}_{\pi_n} \mathbb{1}_{r(q,o)=1} \right| \\
&\leq 2\mathrm{TV}(\tilde{\pi}_n||\pi_n) \\
&\leq 2\sum_{i=1}^{n} \delta_i + \mathrm{TV}(\tilde{\pi}_0, \pi_0) \\
&= 2\sum_{i=1}^{n} \delta_i.
\end{aligned}$$

Assume the sequence $p_n$ converges to $p^*$ the fixed point of $h_{\varepsilon, p_{\mathrm{ref}}}$. Under Assumption 1 we have :

$$\lim_{n \to \infty} |\tilde{p}_n - p_n| \leq 2 \lim_{n \to \infty} \sum_{i=1}^{n} \delta_i = 2\delta^*$$

$\square$

# I  ROBUSTNESS TO NOISE

Fix a query $q$ and let $o \sim \pi_\nu(o \mid q)$. Let $r(q,o) \in \{0,1\}$ be the reward and define

$$p_\nu := \mathbb{E}_{o \sim \pi_\nu(\cdot|q)}\left[\mathbf{1}_{\{r(q,o)=1\}}\right] = \mathbb{P}(r(q,o) = 1 \mid q).$$

Define the flipped label $\tilde{r}$ by

$$\tilde{r} = \begin{cases} r, & \text{with probability } \lambda, \\ 1-r, & \text{with probability } 1-\lambda, \end{cases} \qquad \lambda \in (0,1).$$

**Lemma 4.** *Let $p_\nu := p_\nu(q)$ be the probability of success under $r$ of a policy $\pi_\nu(\cdot|q)$, the noisy reward $\tilde{r}(\cdot, q)$ under $\pi_\nu(\cdot|q)$ is Bernoulli distributed with parameter*

$$\tilde{p}_\lambda(p_\nu) = \mathbb{P}_{\pi_\nu}(\tilde{r}(q,o) = 1 \mid q) = \lambda p_\nu + (1-\lambda)(1-p_\nu) = (2\lambda-1)p_\nu + (1-\lambda).$$

GRPO with a noisy reward $\tilde{r}$ and an anchor $\pi_\circ$ can be therefore written using Theorem 8 as follows:

$$\max_\pi \mathbb{E}_{\pi(\cdot|q)} \tilde{A}_\nu(\cdot, q) - \beta \mathsf{KL}(\pi \| \pi_\circ)$$

where

$$\tilde{A}_\nu(q,o) = \begin{cases} +\omega^+(\tilde{p}_\lambda(p_\nu)), & \tilde{r}(q,o) = 1, \\ -\omega^-(\tilde{p}_\lambda(p_\nu)), & \tilde{r}(q,o) = 0, \end{cases} \tag{26}$$

since $\mathbb{P}_{\nu(\cdot|q)}(\tilde{r}(q,\cdot) = 1) = \tilde{p}_\lambda(p_\nu)$. Let $\Omega(p) = \omega^+(p) + \omega^-(p)$.

Note $p_\circ(q) = p_{\pi_\circ}(q)$ the PoS of $r$ under $\pi_\circ$. Hence the optimal policy is given using again Theorem 8 :

$$\pi^*(o|q) = \frac{\pi_\circ(o|q) \exp \frac{1}{\beta} \tilde{A}_\nu(q,o)}{\tilde{p}_\lambda(p_\circ(q)) \exp(\frac{1}{\beta}\omega^+(\tilde{p}_\lambda(p_\nu(q)))) + (1 - \tilde{p}_\lambda(p_\circ(q))) \exp(-\frac{1}{\beta}\omega^-(\tilde{p}_\lambda(p_\nu(q))))} \tag{27}$$

$$= \frac{\pi_\circ(o|q) \exp\left(\frac{1}{\beta}\left(\omega^+(\tilde{p}_\lambda(p_\nu(q)))\mathbb{1}_{\tilde{r}(q,o)=1} - \omega^-(\tilde{p}_\lambda(p_\nu(q)))\mathbb{1}_{\tilde{r}(q,o)=0}\right)\right)}{\tilde{p}_\lambda(p_\circ(q)) \exp(\frac{1}{\beta}\omega^+(\tilde{p}_\lambda(p_\nu(q)))) + (1 - \tilde{p}_\lambda(p_\circ(q))) \exp(-\frac{1}{\beta}\omega^-(\tilde{p}_\lambda(p_\nu(q))))}. \tag{28}$$

Let $\tilde{Z}_\lambda(q) = \tilde{p}_\lambda(p_\circ(q)) \exp(\frac{1}{\beta}\omega^+(\tilde{p}_\lambda(p_\nu(q)))) + (1 - \tilde{p}_\lambda(p_\circ(q))) \exp(-\frac{1}{\beta}\omega^-(\tilde{p}_\lambda(p_\nu(q))))$

**Lemma 5.** *(PoS under $r$ of Noisy GRPO) The following holds :*

$$p_{\pi^*}(q) = \mathbb{P}_{\pi^*}(r(q,o) = 1) = \frac{1}{\tilde{Z}_\lambda(q)} p_\circ(q) \left[\lambda e^{\frac{1}{\beta}\omega_+(\tilde{p}_\lambda(p_\nu(q)))} + (1-\lambda)e^{-\frac{1}{\beta}\omega_-(\tilde{p}_\lambda(p_\nu(q)))}\right]. \tag{29}$$

*Proof.* Consider the expectation

$$p_{\pi^*}(q) = \frac{1}{\tilde{Z}_\lambda(q)} \mathbb{E}_{\pi_\circ}\left[\mathbf{1}_{\{r(q,o)=1\}} \exp(\frac{1}{\beta}\omega_+(\tilde{p}_\nu)\mathbf{1}_{\{\tilde{r}(q,o)=1\}}) \exp(-\frac{1}{\beta}\omega_-(\tilde{p}_\nu)\mathbf{1}_{\{\tilde{r}(q,o)=0\}})\right],$$

where $\tilde{p}_\nu := \tilde{p}_\lambda(p_\nu(q))$. We drop in the following $q$.

Conditioning on $r$:

$$\mathbb{E}[\cdot] = \mathbb{E}\big[\mathbb{E}[\cdot \mid r]\big].$$

If $r = 0$ the indicator $\mathbf{1}_{\{r=1\}}$ is zero. If $r = 1$, then

$$\mathbb{P}(\tilde{r} = 1 \mid r = 1) = \lambda, \qquad \mathbb{P}(\tilde{r} = 0 \mid r = 1) = 1 - \lambda,$$

so

$$\mathbb{E}\left[\exp\left(\frac{1}{\beta}\omega_+(\tilde{p}_\nu)\mathbf{1}_{\{\tilde{r}=1\}}\right) \exp\left(-\frac{1}{\beta}\omega_-(\tilde{p}_\nu)\mathbf{1}_{\{\tilde{r}=0\}}\right) \mid r = 1\right] = \lambda e^{\frac{1}{\beta}\omega_+(\tilde{p}_\nu)} + (1-\lambda)e^{-\frac{1}{\beta}\omega_-(\tilde{p}_\nu)}.$$

Therefore

$$\mathbb{E}_{\pi_\circ}\left[\mathbf{1}_{\{r(q,o)=1\}} e^{\frac{1}{\beta}\omega_+(\tilde{p}_\nu)\mathbf{1}_{\{\tilde{r}(q,o)=1\}}} e^{-\frac{1}{\beta}\omega_-(\tilde{p}_\nu)\mathbf{1}_{\{\tilde{r}(q,o)=0\}}}\right] = p_\circ\left[\lambda e^{\frac{1}{\beta}\omega_+(\tilde{p}_\nu)} + (1-\lambda)e^{-\frac{1}{\beta}\omega_-(\tilde{p}_\nu)}\right].$$

$\square$

## I.1 Logit Expression

Set

$$a := e^{\frac{1}{\beta}\omega_+(\tilde{p}_\nu)} > 0, \qquad b := e^{-\frac{1}{\beta}\omega_-(\tilde{p}_\nu)} > 0.$$

Then equation 29 becomes

$$p^* = \frac{p_\circ\big(\lambda a + (1-\lambda)b\big)}{\tilde{p}_\lambda(p_\circ)a + \big(1 - \tilde{p}_\lambda(p_\circ)\big)b}.$$

We now compute $p^*/(1-p^*)$. From the expression above,

$$\frac{p^*}{1-p^*} = \frac{p_\circ\big(\lambda a + (1-\lambda)b\big)}{\tilde{p}_\lambda(p_\circ)a + (1 - \tilde{p}_\lambda(p_\circ))b - p_\circ\big(\lambda a + (1-\lambda)b\big)}.$$

The denominator simplifies as

$$\tilde{p}_\lambda(p_\circ)a + (1 - \tilde{p}_\lambda(p_\circ))b - p_\circ\big(\lambda a + (1-\lambda)b\big) = \big(\tilde{p}_\lambda(p_\circ) - \lambda p_\circ\big)a + \big(1 - \tilde{p}_\lambda(p_\circ) - (1-\lambda)p_\circ\big)b$$
$$= (1-\lambda)(1-p_\circ)a + \lambda(1-p_\circ)b$$
$$= (1-p_\circ)\big[(1-\lambda)a + \lambda b\big].$$

Hence

$$\frac{p^*}{1-p^*} = \frac{p_\circ}{1-p_\circ}\frac{\lambda a + (1-\lambda)b}{(1-\lambda)a + \lambda b}. \tag{30}$$

Now rewrite the second factor in terms of $\Omega(\tilde{p}_\nu)$. Since

$$a = e^{\frac{1}{\beta}\omega_+(\tilde{p}_\nu)}, \quad b = e^{-\frac{1}{\beta}\omega_-(\tilde{p}_\nu)}, \quad \Rightarrow \quad a\,e^{\frac{1}{\beta}\omega_-(\tilde{p}_\nu)} = e^{\frac{1}{\beta}\Omega(\tilde{p}_\nu)}, \quad b\,e^{\frac{1}{\beta}\omega_-(\tilde{p}_\nu)} = 1,$$

we get

$$\frac{\lambda a + (1-\lambda)b}{(1-\lambda)a + \lambda b} = \frac{\lambda e^{\frac{1}{\beta}\Omega(\tilde{p}_\nu)} + 1 - \lambda}{(1-\lambda)e^{\frac{1}{\beta}\Omega(\tilde{p}_\nu)} + \lambda}.$$

Let us introduce

$$\Phi_\lambda(\Omega) := \log\frac{\lambda e^\Omega + 1 - \lambda}{(1-\lambda)e^\Omega + \lambda},$$

Taking logs in equation 30 gives

$$\mathrm{logit}(p^*) = \log\frac{p^*}{1-p^*} = \log\frac{p_\circ}{1-p_\circ} + \Phi_\lambda\big(\tfrac{1}{\beta}\Omega(\tilde{p}_\nu)\big).$$

Thus

$$\boxed{\mathrm{logit}(p^*) = \mathrm{logit}(p_\circ) + \Phi_\lambda\big(\tfrac{1}{\beta}\Omega(\tilde{p}_\nu)\big), \qquad \Phi_\lambda(\Omega) := \log\frac{\lambda e^\Omega + 1 - \lambda}{(1-\lambda)e^\Omega + \lambda}.} \tag{31}$$

## I.2 Mirror GRPO iteration with Noisy Reward

We now consider an iteration $(p_n)_{n\geq 0}$ defined by

$$p_n = p_n^*,$$

where

$$p_\circ = p_{n-1}, \qquad \tilde{p}_\nu = \tilde{p}_\lambda(p_{n-1}) = (2\lambda - 1)p_{n-1} + (1-\lambda),$$

and $p_n^*$ is obtained from equation 31 with these choices.

Let

$$L_n := \mathrm{logit}(p_n), \qquad L_{n-1} := \mathrm{logit}(p_{n-1}).$$

Applying equation 31 with $(p_\circ, p_\nu) = (p_{n-1}, \tilde{p}_\lambda(p_{n-1}))$ gives:

$$\boxed{L_n = L_{n-1} + \Phi_\lambda\left(\frac{1}{\beta}\Omega(\tilde{p}_\lambda(p_{n-1}))\right),} \tag{32}$$

with $\Phi_\lambda$ as above.

This iteration will increase the logits at each iteration depending on the sign on $\Phi_\lambda$.

Note that in our case

$$\frac{1}{\beta}\Omega > 0.$$

Then $e^{\frac{1}{\beta}\Omega(.)} > 1$ , and we can determine the sign of the increment $\Phi_\lambda(\frac{1}{\beta}\Omega(,))$.

$$\frac{\lambda e^{\frac{1}{\beta}\Omega} + 1 - \lambda}{(1-\lambda)e^{\frac{1}{\beta}\Omega} + \lambda} - 1 = \frac{(2\lambda - 1)(e^{\frac{1}{\beta}\Omega} - 1)}{(1-\lambda)e^{\frac{1}{\beta}\Omega} + \lambda},$$

whose denominator is strictly positive. Therefore

$$\operatorname{sign}\left(\Phi_\lambda\left(\frac{1}{\beta}\Omega\right)\right) = \operatorname{sign}(2\lambda - 1)$$

Hence for $\lambda > \frac{1}{2}$ logit increases and for $\lambda < \frac{1}{2}$ it decreases.

For Mixed Ref-Mirror GRPO ($\alpha > 0$ in equation 9), the anchor $p_\circ = \tilde{p}_{n-1}^{(\alpha)}$ defined in Lemma 3 that incorporates the reference probability of success $p_{\text{ref}}$. For $\lambda = 1$, we recover GRPO iterations in the noiseless case derived earlier.

## J   PLOTS

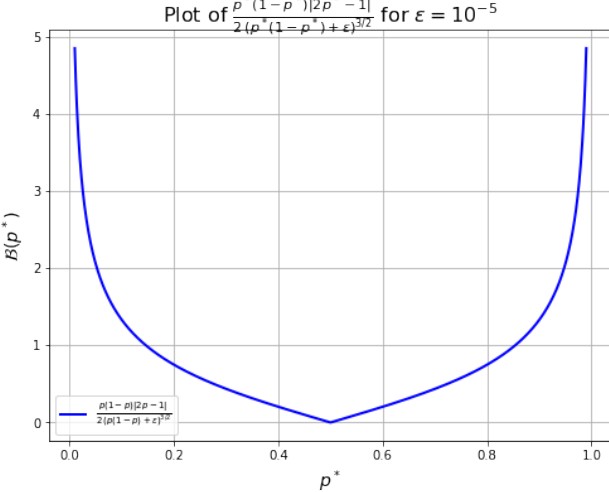

Figure 9: Lower bound on $\beta$ to ensure local convergence of GRPO fixed point iteration.

## K   ASSETS

**Hardware setup**   All our experiments were run on one compute node with Dual 48-core Intel Xeon 8468, 2TB of RAM, 8 NVIDIA HGX H100 80GB SMX5, 8x 3.4TB Enterprise NVMe U.2 Gen4, and 10x NVIDIA Mellanox Infiniband Single port NDR adapters, running RedHat Enterprise Linux 9.5

**GRPO Config Setup**   We use the group size $G = 16$ and per-device batch size 16 meaning each on each GPU a single prompt $x$ with 16 corresponding responses is processed. To increase the overall batchsize we use gradient accumulation of 4, ending with an effective batch size of prompts of 28. The context length used for this experiment is 200, and the sampling temperature is set to $\tau = 0.1$.

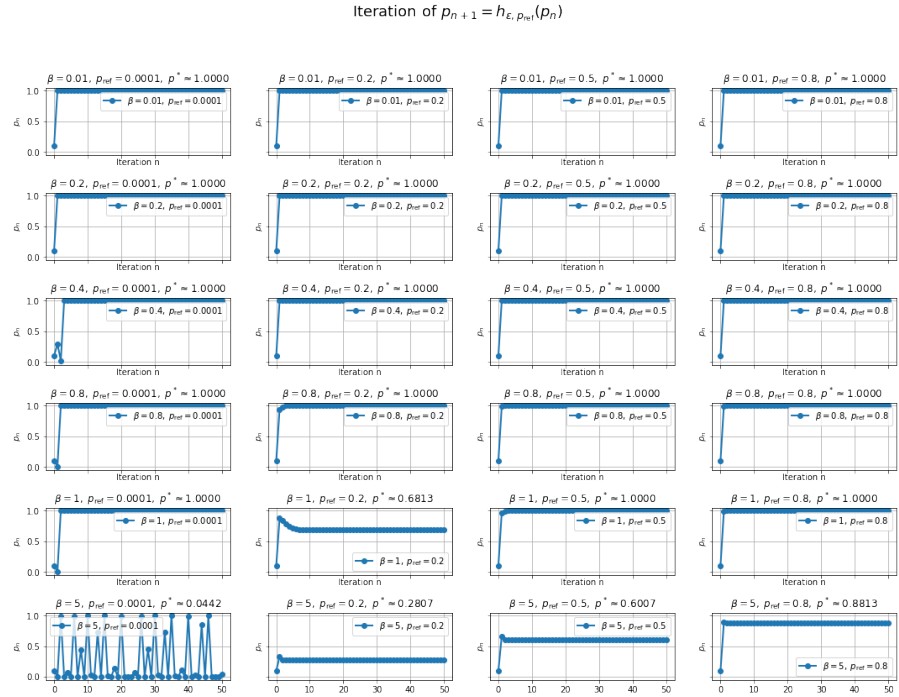

Figure 10: GRPO Recursion and convergence to fixed points of $h_\varepsilon$, for $\varepsilon = 1e^{-5}$

**Libraries** Our experiments rely on the open-source libraries pytorch (Paszke et al., 2019) (license: BSD), HuggingFace Transformers (Wolf et al., 2020) (Apache 2.0 license), and HuggingFace TRL (von Werra et al., 2020a) (Apache 2.0 license).

