# OpenReview forum: "Reinforcement Learning with Verifiable Rewards: GRPO's Loss, Dynamics, and Success Amplification"
_ICLR.cc/2026/Conference — Submitted to ICLR 2026_

### Official Review · Reviewer_82Wa · 2025-10-27

**Soundness:** 2
**Presentation:** 2
**Contribution:** 2
**Rating:** 4
**Confidence:** 2

**Summary:**

This paper focuses on Reinforcement Learning with Verifiable Rewards and provides a theoretical analysis of several key properties of GRPO.

**Strengths:**

This paper provides a theoretical explanation of several key properties of GRPO and presents several recommendations for its use.

**Weaknesses:**

1. The abstract should not include citations.
2. The related work analysis should be moved to a separate "Related Work" section.
3. The structure of the abstract and introduction is somewhat unclear, making it difficult to identify the paper's motivation, i.e., the challenge it aims to address.
4. Important experimental results should be included in the main text.
5. Besides GRPO, are the theoretical results in this paper generalizable? Specifically, can they be applied to other SOTA RL methods in LLMs? If so, further experiments on additional RL methods should be conducted.

**Questions:**

Besides GRPO, are the theoretical results in this paper generalizable? Specifically, can they be applied to other SOTA RL methods in LLMs? If so, further experiments on additional RL methods should be conducted.

---

> ### Author Response · Authors · 2025-11-22
> **Response to Reviewer 82Wa**
>
> We thank the reviewer for their valuable comments and suggestions, we updated the paper all edits are in blue.
>
> ___
> **The abstract should not include citations.**
> **The related work analysis should be moved to a separate "Related Work" section.**
> **The structure of the abstract and introduction is somewhat unclear, making it difficult to identify the paper's motivation, i.e., the challenge it aims to address.**
>
> We addressed all these, removed citation from abstract, labeled related works within introduction. Made changes to the introduction flow to highlight the motivation and the goal of the paper.
> ___
> **Important experimental results should be included in the main text.**
>
> We moved all experiments to main paper and added exhaustive experimentations and ablations in Section 7 page 9-10 in the updated paper that compares all variant of GRPO : clipped with mean-var normalization and KL to reference; DR GRPO clipped with mean normalization with KL to reference; Mirror GRPO (mean and mean-var normalization); Mixed Ref Mirror GRPO (mean and var-norm). DAPO (clipping , mean-var , no KL to reference).
>
> We did our experimentation and compared all these variants in the noiseless and the noisy setting as suggested (large and moderate noise levels).
>
> We confirmed that clipped versions are on par with mirror variants in the noiseless setting in terms of PoS amplification, with DAPO outperforming all variants after hyper parameter tuning in the noiseless setting .
>
> We found that original GRPO (clipped,mean-var,ref) and Mixed Ref Mirror GRPO are robust in moderate noise levels. Variants that do not incorporate regularization to the reference models such as DAPO are not robust in the moderate regime. All variants deteriorate in high noise regime.
>
> Additional plots are shown in Appendix D page 14/15/16 in the updated paper.
>
> Note that in Section 6 (page 9 ) we study the dynamics of the PoS under noisy rewards with all proof included in Appendix I (pages 24/25/26), proving that GRPO has some robustness for moderate noise levels and that it deteriorates in high noise regimes.
>
> ___
> **Besides GRPO, are the theoretical results in this paper generalizable? Specifically, can they be applied to other SOTA RL methods in LLMs? If so, further experiments on additional RL methods should be conducted.**
>
> The results in the paper cover GRPO and its variants , they don't cover PPO for instance. We presented in the updated paper exhaustive experiments on GRPO and its variants (DAPO, Dr GRPO, Mirror, Mixed Mirror ) in the noiseless and the noisy setting.
>
> We hope those additional experiments and new theory on the robustness of GRPO to noise will convince the reviewer of the merits of the work !

---

> > ### Author Response · Authors · 2025-11-26
> > **Rebuttal**
> >
> > Dear Reviewer,
> >
> > Thank you for  reviewing our submission. We have posted a rebuttal  and a paper revision addressing all your comments regarding additional experiments and robustness to noise.
> >
> > With the rebuttal phase ending soon, we kindly ask whether you could take our responses into account and let us know if this clarifies the questions and concerns you had.
> >
> > Best,
> >
> > Authors

---

### Official Review · Reviewer_eWXX · 2025-10-27

**Soundness:** 3
**Presentation:** 3
**Contribution:** 2
**Rating:** 4
**Confidence:** 1

**Summary:**

This paper provides a theoretical analysis of GRPO for reinforcement learning with verifiable rewards.
It reformulates GRPO as a weighted contrastive loss and derives explicit success-rate dynamics, showing how the algorithm amplifies success probability and converges to a fixed point. The paper further studies variants such as Mirror GRPO and dual-KL GRPO, and experiments on GSM8K validate the predicted success amplification behavior.

**Strengths:**

1. **Clear theoretical formulation.**
The paper offers a clean and rigorous analysis of GRPO, reformulating it as a weighted contrastive learning problem and deriving explicit success-rate dynamics that explain its empirical behavior.

2. **Unifying perspective.**
It connects GRPO to broader RL principles such as mirror descent and KL-regularized optimization, providing a unified interpretation of several GRPO variants (Mirror GRPO, dual-KL GRPO, Dr.GRPO).

**Weaknesses:**

**Limited experimental scope and analysis.**
The experiments are limited to the GSM8K math reasoning dataset with experiements seems unrealted to the  (and place them in the appendix with fewer discussions related to it), leaving it unclear whether the theoretical findings and success amplification behavior generalize to other RLHF or verifiable reward tasks such as dialogue or code generation.

**Questions:**

Could the authors consider moving some of the appendix experiments into the main text and expanding the empirical discussion?
In particular, it would be valuable to include a deeper analysis of how different GRPO variants (e.g., Mirror GRPO, dual-KL GRPO, Dr.GRPO) behave empirically, and to discuss why their performance or convergence patterns differ in practice.

---

> ### Author Response · Authors · 2025-11-22
> **Response to Reviewer eWXX**
>
> Thank you for your valuable comments and suggestions. We updated the paper and all edits are in blue, we integrated all your suggestions.
> ___
> **Limited experimental scope and analysis. The experiments are limited to the GSM8K math reasoning dataset with experiements seems unrealted to the (and place them in the appendix with fewer discussions related to it), leaving it unclear whether the theoretical findings and success amplification behavior generalize to other RLHF or verifiable reward tasks such as dialogue or code generation.**
> **Could the authors consider moving some of the appendix experiments into the main text and expanding the empirical discussion? In particular, it would be valuable to include a deeper analysis of how different GRPO variants (e.g., Mirror GRPO, dual-KL GRPO, Dr.GRPO) behave empirically, and to discuss why their performance or convergence patterns differ in practice.**
>
> We moved all experiments to main paper and added exhaustive experimentations and ablations in Section 7 page 9-10 in the updated paper that compares all variant of GRPO as suggested by the reviewer: clipped with mean-var normalization and KL to reference; DR GRPO clipped with mean normalization with KL to reference; Mirror GRPO (mean and mean-var normalization); Mixed Ref Mirror GRPO (mean and var-norm). DAPO (clipping , mean-var , no KL to reference).
>
> We did our experimentation and compared all these variants in the noiseless and the noisy setting as suggested (large and moderate noise levels).
>
> We confirmed that clipped versions are on par with mirror variants in the noiseless setting in terms of PoS amplification, with DAPO outperforming all variants after hyper parameter tuning in the noiseless setting .
>
> We found that original GRPO (clipped,mean-var,ref) and Mixed Ref Mirror GRPO are robust in moderate noise levels. Variants that do not incorporate regularization to the reference models such as DAPO are not robust in the moderate regime. All variants deteriorate in high noise regime.
>
> Additional plots are shown in Appendix D page 14/15/16 in the updated paper.
>
> Note that in Section 6 (page 9 ) we study the dynamics of the PoS under noisy rewards with all proof included in Appendix I (pages 24/25/26), proving that GRPO has some robustness for moderate noise levels and that it deteriorates in high noise regimes.
>
> We hope those additional experiments and new theory on the robustness of GRPO to noise will convince the reviewer of the merits of the work !

---

> > ### Comment · Reviewer_eWXX · 2025-11-24
> >
> > Thank you for your explanation!
> >
> > However, I would like to note that I am not well qualified to review this submission.
> >
> > My research background is primarily in reinforcement learning theory and control, but I am not sufficiently familiar with large language model (LLM)–related RL works. During the bidding stage, I explicitly marked myself as unwilling to review many RL for LLM papers. However, this submission (as well as some else) was still assigned to me.
> >
> > To ensure the fairness and quality of the review process, I have noted AC and respectfully request that my review be excluded from the final decision, and to seek other reviewers who have stronger expertise in RLHF or LLM fine-tuning.

---

### Official Review · Reviewer_6zCh · 2025-11-01

**Soundness:** 3
**Presentation:** 3
**Contribution:** 2
**Rating:** 6
**Confidence:** 3

**Summary:**

The paper provides a detailed theoretical analysis of GRPO, an RL method used to fine-tune LLMs using verifiable rewards. The authors show that GRPO can be viewed as an adaptive contrastive loss and derive mathematical recursive equations describing how the policy evolves over the course of training. Several variants are analyzed, including standard GRPO, Mirror-GRPO, and Dr. GRPO, with comparison of different normalization and regularization schemes. The results show that GRPO consistently increases the model’s probability of success. The proved theory predicts convergence properties for Mirror-GRPO. Additionally, an experiment on the GSM8K dataset shows that GRPO increases the success rate from 21% to 37.5%.

**Strengths:**

The paper provides mathematical grounding for GRPO, showing that different variants of the algorithm increase the probability of success of the learned policy. This is proven mathematically by the authors, showing that the probability of success converges to a fixed point (and the authors derive closed-form recursions). This offers theoretical insight into how the GRPO algorithm improves on stability and convergence. An experiment, although small, has consistent results with the theoretical predictions. Finally, the authors provide practical takeways that the readers may benefit from.

**Weaknesses:**

- The paper is very theory-focused, with limited experimental validation at small scale and without ablation studies. Besides, having some of these results in the main body of the paper would be benefitial, and more results with the different considered GRPO variants could improve the paper. Since almost any fine-tuning with verifiable rewards can raise the success rate, it's unclear whether the observed improvements come from GRPO itself or standard additional fine-tuning.
- The paper could also benefit from a few intuitive examples of diagrams to illustrate the theory and why the various dynamics/variants help, in some sections, before diving into the equations.
- At the end of the introduction, the paper could benefit from a higher-level explanation of the practical implications of the results shown in the paper; and at the end of the paper, of some mention of limitations, for instance regarding the assumptions made in the theory parts.

**Questions:**

* I am curious, as authors claim having improved success rate from 21% to 37.5% using GRPO, whether they have compared it to different fine-tuning methods such as PPO or DAPO? Do the authors have some idea about how GRPO improvement would translate in terms of sample efficiency or runtime, compared to other fine-tuning algorithms?
* How robust do the authors think the theoretical results would expand in the context of noisy imperfect binary rewards, which can be found in real-world LLM applications? What about continuous reward settings such as human feedback?
* Did the authors test the mirror GRPO version empirically, as the theory predicts a monotonic improvement of the success probability?

---

> ### Author Response · Authors · 2025-11-22
> **Response to Reviewer 6zCh**
>
> We thank the reviewer for their positive feedback and for their great suggestions that we incorporated in the updated paper,  and that have enlarged the scope of the paper. All updates are in blue.
>
> **The paper is very theory-focused, with limited experimental validation at small scale and without ablation studies. Besides, having some of these results in the main body of the paper would be benefitial, and more results with the different considered GRPO variants could improve the paper. Since almost any fine-tuning with verifiable rewards can raise the success rate, it's unclear whether the observed improvements come from GRPO itself or standard additional fine-tuning.**
>
> **I am curious, as authors claim having improved success rate from 21% to 37.5% using GRPO, whether they have compared it to different fine-tuning methods such as PPO or DAPO?**
>
> **Did the authors test the mirror GRPO version empirically, as the theory predicts a monotonic improvement of the success probability?**
>
> **How robust do the authors think the theoretical results would expand in the context of noisy imperfect binary rewards, which can be found in real-world LLM applications?**
> ___
>
> We added exhaustive experimentations and ablations in Section 7 page 9-10 in the updated paper that compares all variant of GRPO as suggested by the reviewer: clipped with mean-var normalization and KL to reference; DR GRPO clipped with mean normalization with KL to reference; Mirror GRPO (mean and mean-var normalization); Mixed Ref Mirror GRPO (mean and var-norm). DAPO (clipping , mean-var , no KL to reference).
>
> We did our experimentation and compared all these variants in the noiseless and the noisy setting as suggested (large and moderate noise levels).
>
> We confirmed that clipped versions are on par with mirror variants in the noiseless setting in terms of PoS amplification, with DAPO outperforming all variants after hyper parameter tuning in the noiseless setting .
>
> We found that original GRPO (clipped,mean-var,ref)  and Mixed Ref Mirror GRPO are robust in moderate noise levels. Variants that do not incorporate regularization to the reference models such as DAPO are not robust in the moderate regime. All variants deteriorate in high noise regime.
>
> Additional plots  are shown in Appendix D page 14/15/16 in the updated paper.
>
> Note that in Section 6 (page 9 ) we study the dynamics of the PoS under noisy rewards with all proof included in Appendix I (pages 24/25/26), proving that GRPO has some robustness for moderate noise levels and that it deteriorates in high noise regimes.
>
> ___
> **What about continuous reward settings such as human feedback?**
>
> **and at the end of the paper, of some mention of limitations, for instance regarding the assumptions made in the theory parts.**
>
> We added a limitation section in page 13. We focused on binary rewards in bandit style updates to perform the analysis of what GRPO learns in this idealized setting. This setting remains interesting for example in code generation or in mathematical reasoning in a single step, where verifiable rewards are available and used in practice. For continuous rewards the analysis does not extend in a straightforward way, one can turn turn continuous to binary via thresholding, but it would be interesting to extend our analysis to that setting, and leave as an open question for future work.
>
> ___
> **Do the authors have some idea about how GRPO improvement would translate in terms of sample efficiency or runtime, compared to other fine-tuning algorithms?**
>
> The main appeal of GRPO is in not having to train the policy and  a value network as in PPO  and it hinges on monte-carlo samples that can be obtained efficiently with inference frameworks such as vLLM.

---

> > ### Author Response · Authors · 2025-11-26
> > **rebuttal**
> >
> > Dear Reviewer,
> >
> > Thank you for  reviewing our submission. We have posted a rebuttal  and a paper revision addressing all your comments regarding additional experiments and robustness to noise.
> >
> > With the rebuttal phase ending soon, we kindly ask whether you could take our responses into account and let us know if this clarifies the questions and concerns you had.
> >
> > Best,
> >
> > Authors

---

### Official Review · Reviewer_tyaa · 2025-11-14

**Soundness:** 3
**Presentation:** 3
**Contribution:** 2
**Rating:** 4
**Confidence:** 3

**Summary:**

The paper provides a formal analysis of GRPO-style RL for verifiable, 0/1 rewards. It derives closed-form policy updates and success-probability (PoS) dynamics for GRPO, Mirror-GRPO, and Dr.GRPO (with and without reference KL), showing that whitened advantages can be understood as an adaptive contrastive objective over successful vs failed trajectories. The main theoretical results characterize fixed points and monotonicity of PoS, with Mirror-GRPO shown to monotonically drive PoS to 1 under idealized assumptions.

**Strengths:**

1. Provides a clean, distribution-level view of GRPO that matches current practice and clarifies the roles of whitening and KL regularization.

2. The theory is tied back to practice with small but sensible experiments that qualitatively follow the predicted PoS amplification behavior.

**Weaknesses:**

1. The setting is highly idealized: single-step, 0/1 verifiable rewards and bandit-style updates, which sidesteps multi-turn reasoning or tool using cases.

2. Lack of experimental-level analysis.

3. The main analysis omits clipping and several stabilizing tricks that matter in large-scale GRPO, so it is unclear how close real training is to the derived dynamics.

4. Empirical evaluation is narrow and mostly illustrative; it does not stress-test where the theory breaks down (noisy or delayed rewards, strong model misspecification, heavy off-policy data).

**Questions:**

See above.

---

> ### Author Response · Authors · 2025-11-22
> **Response to Reviewer tyaa**
>
> We thank the reviewer for their valuable feedback and suggestions. Please note that we uploaded an update paper, all edits are in blue. We address here the main questions:
> ___
> **The setting is highly idealized: single-step, 0/1 verifiable rewards and bandit-style updates, which sidesteps multi-turn reasoning or tool using cases.**
>
> We added a limitation section in page 13. We focused on binary rewards in bandit style updates to perform the analysis of what GRPO learns in this idealized setting.  Please note that in appendix H we give an analysis of how we can leverage our setup to get bound on PoS using a hypothesis class or parametric policies under approximation errors assumptions.
>
> The verifiable reward setting remains interesting for example in code generation or in mathematical reasoning in a single step, where verifiable rewards are available and used in practice.
> ___
>
> **Lack of experimental-level analysis.**
> **Empirical evaluation is narrow and mostly illustrative; it does not stress-test where the theory breaks down (noisy or delayed rewards, strong model misspecification, heavy off-policy data).**
>
> We added exhaustive experimentations and ablations in Section 7 page 9-10 in the updated paper that compares all variant of GRPO: clipped with mean-var normalization and KL to reference;  DR GRPO clipped with mean normalization with KL to reference; Mirror GRPO (mean and mean-var normalization); Mixed Ref Mirror GRPO (mean and var-norm). DAPO (clipping , mean-var , no KL to reference).
>
> We did our experimentation and compared all these variants in the noiseless and the noisy setting (large and moderate noise levels).
> We confirmed that clipped versions are on par with mirror variants in the noiseless setting in terms of PoS amplification.  And found that original GRPO and Mixed Ref Mirror GRPO  are robust in moderate noise levels. Variants that do not incorporate regularization to the reference models such as DAPO are not robust in the moderate regime. All variants deteriorate in high noise regime.
>
> Additional experimental results are shown in Appendix D page 14/15/16 in the updated paper.
>
> Note that in Section 6 (page 9 ) we study the dynamics of the PoS under noisy rewards with all proof included in Appendix I (pages 24/25/26), proving that GRPO has some robustness for moderate noise levels and that it deteriorates in high noise regimes.
> ___
>
> **The main analysis omits clipping and several stabilizing tricks that matter in large-scale GRPO, so it is unclear how close real training is to the derived dynamics.**
>
> Clipping was introduced as an approximation the mirror regularizer  (KL to previous policy) in the PPO paper, this was motivated by the difficulty of computing the KL for continuous policies. In the case of LLMs we can compute the KL and use Mirror GRPO. Our experiments showed that Mirror variants outperforms clipping variants except for DAPO that performs well in the noiseless regime.
> ___
>
> **Empirical evaluation is narrow and mostly illustrative; it does not stress-test where the theory breaks down (noisy or delayed rewards, strong model misspecification, heavy off-policy data).**
>
> As mentioned above we added exhaustive experiments and studied theoretically and empirically the robustness of  GRPO to noise, stress testing and proving when the dynamics break down in the noisy regime and when they have a certain degree of robustness.  Please note that in Section 6 (page 9 ) we added a theoretical analysis on the dynamics of the PoS under noisy rewards with all proof included in Appendix I (pages 24/25/26), proving that GRPO has some robustness for moderate noise levels and that it deteriorates in high noise regimes.
>
> ___
>
> We thank you for your suggestions that enlarged the scope of the paper !

---

> > ### Author Response · Authors · 2025-11-26
> > **rebuttal**
> >
> > Dear Reviewer,
> >
> > Thank you for  reviewing our submission. We have posted a rebuttal  and a paper revision addressing all your comments regarding additional experiments and stress testing the setup and robustness to noise.
> >
> > With the rebuttal phase ending soon, we kindly ask whether you could take our responses into account and let us know if this clarifies the questions and concerns you had.
> >
> > Best,
> >
> > Authors

---

### Author Response · Authors · 2025-11-22
**Common comment to all reviewers**

We thank all the reviewers for their feedback. We uploaded a new version of the papers with all edits in the blue:

Main additions:
*  **Section 6 (page 9 ): GRPO Robustness to Noisy Reward Theory** we study the dynamics of the PoS under noisy rewards with all proofs included in Appendix I (pages 24/25/26), proving that GRPO and  its variants  have some robustness for moderate noise levels and that it deteriorates in high noise regimes.
*  **Section 7 page 9-10: Experimental Validation (Noiseless and Noisy settings)** in the updated paper that compares all variant of GRPO: clipped with mean-var normalization and KL to reference; DR GRPO clipped with mean normalization with KL to reference; Mirror GRPO (mean and mean-var normalization); Mixed Ref Mirror GRPO (mean and var-norm). DAPO (clipping , mean-var , no KL to reference).
* Additional experimental results are shown in Appendix D page 14/15/16 in the updated paper.

Minor:
* updates to abstract, introduction, added limitation section, edited conclusion on robustness

---

### Meta-Review · Area_Chair_AmpY · 2025-12-08

**Summary:**

Overall, I find that this paper offers a clean and unified theoretical view of GRPO under verifiable binary rewards, but that its contribution is somewhat limited in depth and scope. The setting is highly idealized (single-step, 0/1 rewards, no tools or long-horizon effects), and the empirical support is modest, using small models and narrow benchmarks. In light of these factors, I believe the work falls slightly below the bar for ICLR main conference and recommend rejection, while acknowledging its value as a clarifying analysis for practitioners working specifically on GRPO-style RLVR. I encourage the author to resubmit to another venue after improving the writing significantly.

**Reviewer Concerns:**

The rebuttal did a good job of addressing some specific technical concerns, in particular by adding noisy-reward analysis, clarifying the interpretation of whitening as an adaptive KL strength, and expanding the experimental section to compare several GRPO variants. However, the core outstanding issues raised by multiple reviewers remain: the theoretical results are derived in a very simplified bandit-style setting, the gap to realistic large-scale RL for LLMs is substantial, and the empirical evaluation is still too limited to convincingly demonstrate practical impact. As a result, I agree with the reviewers who view the work as insightful but not yet strong enough in either theory depth or empirical validation for acceptance.

**Reviewer Scores:**

I believe R1’s (score 4) main reservations about idealization and limited empirical support would largely remain after the rebuttal, so their score would likely stay at 4. R2, who was the most positive (score 6), had many of their technical concerns addressed and would probably keep their score at 6, but not raise it given the remaining scope/impact issues. R3 explicitly requested recusal due to low expertise in this area, so I do not weigh that review; R4’s concerns were mainly about presentation and scope, which have been partially mitigated by revisions, but I still expect their overall assessment (around 4) would remain unchanged.

---

### Decision · Program_Chairs · 2026-01-26

Reject